# Aligning With Human Values Without Revealing Human Judgements

## Abstract

With the increasing ubiquity of large language models it has become crucial to ensure guarantees for models trained to be aligned with human values to avoid leaking information on the human judgements that have been provided to the algorithm. To target this issue we focus on the problem of alignment via reinforcement learning from human preference rankings, subject to the constraint of not revealing any information on the human data used to align the model. To achieve this, we analyze $(\epsilon, \delta)$-DP for both the Bradley-Terry-Luce (BTL) model and the Plackett-Luce (PL) model. We introduce a theoretically founded algorithm for learning rewards from human rankings that achieves this objective without leaking the human rankings. We further demonstrate that the privately learned rewards can be used to train policies achieving statistical performance guarantees that asymptotically match the best known algorithms in the non-private setting, which are in some cases minimax optimal. Strikingly, our analysis and our results reveal that it is possible to obtain the same model performance without any trade-off on the protection of the human judgments, and our paper provides the first algorithms that can achieve provable privacy of human judgements, while still producing aligned models with optimal performance.

## 1 Introduction

With the rise of large pretrained machine learning models that flexibly interact with humans, there is an increasing need to ensure that the models do not exhibit harmful behaviour or ethical violations that can cause unsafe circumstances for humans. Reinforcement Learning from Human Feedback (RLHF) is currently the standard method targeting this problem (OpenAI, 2023; Google Gemini, 2023), and has achieved significant success introducing several behavioral skills to language models (i.e. probability distributions over sequences of tokens (Shannon, 1948)) from refusing to act on improper requests to simply interacting with humans by responding to questions (Ziegler et al., 2019; Wu et al., 2021; Nakano et al., 2021; Stiennon et al., 2020; Abramson et al., 2022; Glaese et al., 2022; Bai et al., 2022; Ganguli et al., 2022; Menick et al., 2022; Ouyang et al., 2022; Gao et al., 2023; Ramamurthy et al., 2023). Yet there are still problems with large language models where recent work demonstrates the unethical behaviour that they can exhibit when they interact with humans (Ganguli et al., 2022; Perez et al., 2022).

While improving the safety and harmlessness of LLMs remains an active area of research, the use of RLHF introduces an orthogonal set of problems relating to human interactions. In particular, the input data used for RLHF training is human ratings of model responses to prompts. Furthermore, current language models record data when interacting with humans via chat interfaces, and this data can be used for future training (OpenAI, 2023). As a result, there are numerous reasons to worry about privacy when building a reward function from human-feedback, a few of which we now enumerate. First, even when human raters are paid, they may be giving preference ratings that need to be kept private. Honest preference ratings on sensitive topics can be very revealing of private information e.g. political preference, gender identity etc. In some jurisdictions there is a legal mandate for an employer of paid raters to avoid leaking such private information. Moreover, orthogonal to this, several other issues would arise for an LLM trained to give basic medical advice based on responses from raters who have real medical issues, in which case the preference data clearly must be kept private. Furthermore, a glance at the terms of use of the major LLM providers indicates that user feedback on LLM outputs is being stored, with the possibility of later training on this data.

Thus, the current paradigm of only using paid raters may change in the foreseeable future. From another perspective, human preference data on specialized topics, e.g. legal or scientific questions, may be very expensive to obtain, and thus could be viewed as trade secrets that must be kept private. In this case, privacy is economically incentivized for providers of LLMs. Furthermore, one of the major LLM providers has issued a statement indicating that maintaining privacy is a core principle of responsible AI development DeepMind (2023). Finally, the ability to violate privacy with only access to anonymous preference rankings was conclusively demonstrated in Narayanan and Shmatikov (2008), for the case of the Netflix prize dataset consisting of anonymous movie ratings by Netflix users. This paper shows that it is possible to deanonymize a target user with just a small amount of publicly available information about the target, and then subsequently to learn potentially sensitive information about the user e.g. political preferences. Therefore, it seems quite plausible that access to an LLM fine-tuned on human preference data, combined with the above well-established methods for deanonymization from preference data alone, can lead directly to privacy violations.

Thus, as large language models continue to scale to interact with millions of people in more complex ways, the necessity of maintaining the privacy of individual interactions becomes even more significant. To mitigate the privacy risks associated with machine learning, the framework of differential privacy is the primary approach to the design of algorithms with rigorous privacy guarantees (Dwork et al., 2006; Dwork and Roth, 2014).

The standard approach to RLHF starts with a pretrained language model and fixed dataset of prompts. A prompt is sampled from the dataset, and $K$ outputs from the language model are sampled conditioned on the prompt. A human rater then gives a preference ranking of the $K$ outputs. This process is repeated until a dataset $\mathcal{D}$ containing $n$ human preference rankings over model responses is collected. Following this a reward model $r_\theta$ is trained to match the human preference rankings in $\mathcal{D}$. Finally, the original pretrained language model is further trained via reinforcement learning to maximize the learned rewards $r_\theta$. Both the human ratings and prompts in the dataset $\mathcal{D}$ are generated by humans interacting with the model, and thus may contain information that should be kept private even when the final trained model is released to the public.

Recent work of Zhu et al. (2023) studies the sample complexity of RLHF, and gives an algorithm achieving minimax optimal rates for RLHF in the setting where rewards are linearly parametrized in some feature space. In this paper we will prove that, in the same setting, it is possible to achieve minimax optimal sample complexity and differential privacy simultaneously. In particular, our differential privacy guarantees imply that even if $n-1$ of the human ratings in the dataset are revealed, it will be statistically infeasible to learn the private information of the one remaining rating, when given access to the final trained model.

## 1.1 OUR RESULTS

We begin by introducing the basic setting for RLHF. There are a set of states $S$ and actions $A$ corresponding to prompts and language model responses respectively. First a state $s$ is sampled from a distribution $\rho$, then $K$ actions $a_1, \ldots, a_K$ are sampled from the model conditioned on the state $s$ giving a tuple $(s, a_1, \ldots, a_K)$. Human preference rankings over $a_1, \ldots, a_K$ are given by a permutation $\sigma : [K] \to [K]$, where $a_{\sigma(1)}$ is the most preferred action, and $a_{\sigma(K)}$ is the least preferred action. We assume that there is a feature map $\phi : S \times A \to \mathbb{R}^d$, and a reward modelled as a linear function $r_\theta(s, a) = \langle \theta, \phi(s, a) \rangle$. Human preference rankings over model responses are assumed to follow a Plackett-Luce model (Plackett, 1975; Luce, 2012) for some true reward $r_{\theta^*}$. That is the probability that an action $a_i$ is selected as the "best" from a list of $K$ alternative actions is proportional to

$$\mathbb{P}[a_i | s, a_1, \ldots a_K] = \frac{\exp(r_{\theta^*}(s, a_i))}{\sum_{j=1}^{K} \exp(r_{\theta^*}(s, a_j))}.$$

This naturally implies a distribution on full rankings of actions $\sigma : [K] \to [K]$, by first selecting the best action from the full list of $K$ actions, then recursively selecting the next best from the remaining $K-1$, and so on. When $K = 2$ this is equivalent to the Bradley-Terry-Luce model. We denote by $\mathcal{D}$ the dataset of $n$ human ranking tuples $(s, a_1, \ldots a_K, \sigma)$. In order to accurately estimate uncertainty in the rewards given the dataset $\mathcal{D}$, one typically uses the dataset-dependent covariance matrix given

by

$$\Sigma_D = \frac{2}{nK(K-1)} \sum_{i=1}^{n} \sum_{j=1}^{K} \sum_{j=k+1}^{K} \left( (\phi(s^i, a_j^i) - \phi(s^i, a_k^i))(\phi(s^i, a_j^i) - \phi(s^i, a_k^i))^\top \right).$$

In particular, the pessimistic policy optimization algorithm in our paper (as well as in Zhu et al. (2023)) depends on access to a sufficiently accurate approximation of $\Sigma_\mathcal{D}$.

**RLHF for Contextual Bandits.** Our first results are in the contextual bandit setting, where states $s$ are sampled from some fixed distribution $\rho$. This is the closest to the standard setup of RLHF applied to LLM alignment. Under certain regularity assumptions the results of Zhu et al. (2023) show that computing the maximum likelihood estimator (MLE) $\hat{\theta}_{\text{MLE}}$ for the reward parameters, followed by a pessimistic policy optimization algorithm with respect to $r_{\hat{\theta}_{\text{MLE}}}$ yields a policy $\hat{\pi}_{\text{PE}}$ achieving expected rewards that are at most $O\left(\sqrt{\frac{d}{n}}\right)$ worse than those of the optimal policy. Our main result matches this performance while simultaneously achieving differential privacy for the dataset $\mathcal{D}$.

**Theorem 1.1.** *(Informal) Let $\mathcal{D}$ be a dataset of $K$-wise human rankings of the form $(s, a_1, \ldots a_k, \sigma)$. Under appropriate regularity assumptions, there is an $(\epsilon, \delta)$-differentially private algorithm that learns a reward model $r_{\widetilde{\theta}_{\text{MLE}}}$ and a perturbed data covariance $\widetilde{\Sigma}_\mathcal{D}$ from $\mathcal{D}$. Both $\widetilde{\theta}_{\text{MLE}}$ and $\widetilde{\Sigma}_\mathcal{D}$ are close (under appropriate metrics) to the true parameter $\theta^*$ and the true data covariance $\Sigma_\mathcal{D}$ respectively.*

**Theorem 1.2.** *(Informal) Under appropriate regularity assumptions, there is pessimistic policy optimization algorithm that, when trained with the reward model $r_{\widetilde{\theta}_{\text{MLE}}}$ and data covariance estimate $\widetilde{\Sigma}_\mathcal{D}$ outputs a policy $\widetilde{\pi}_{PE}$ achieving rewards that are worse than the optimal policy by at most*

$$O\left(\sqrt{\frac{d}{n}} + \frac{(d \log(1/\delta))^{1/4}}{\sqrt{\epsilon n}}\right)$$

In the typical differential privacy setting $\epsilon$ is a constant and $\delta$ is inverse polynomial in $n$, and so the first term above dominates. Thus, in the typical setting our results match the minimax optimal rate $O\left(\sqrt{\frac{d}{n}}\right)$ up to constant factors. Also notable in our results is the fact that privacy holds for the estimated reward function $r_{\widetilde{\theta}_{\text{MLE}}}$ and the perturbed data covariance $\widetilde{\Sigma}_\mathcal{D}$. This makes our results modular, and means that privacy will be maintained under follow-up post-processing by any policy learning algorithm. In particular, it is even possible to publicly release the weights $\widetilde{\theta}_{\text{MLE}}$ of the learned reward model $r_{\widetilde{\theta}_{\text{MLE}}}$, along with the perturbed data covariance $\widetilde{\Sigma}_\mathcal{D}$.

**RLHF for general MDPs.** We extend our results to RLHF in general MDPs, where human preferences are given over pairs of trajectories. In this setting we also simultaneously obtain $(\epsilon, \delta)$-differential privacy and performance matching the non-private algorithm.

**Theorem 1.3.** *(Informal) Let $\mathcal{D}_\tau$ be a dataset of pairwise trajectory comparisons from an MDP $M$. Under appropriate regularity assumptions, there is an $(\epsilon, \delta)$-differentially private algorithm that learns a reward model $r_{\widetilde{\theta}_{\text{MLE}_\tau}}$ and a perturbed data covariance $\widetilde{\Sigma}_{\mathcal{D}_\tau}$ from $\mathcal{D}_\tau$. Both $\widetilde{\theta}_{\text{MLE}_\tau}$ and $\widetilde{\Sigma}_{\mathcal{D}_\tau}$ are close in an appropriate metric to the true parameter $\theta^*$ and the true data covariance $\Sigma_{\mathcal{D}_\tau}$ respectively.*

**Theorem 1.4.** *(Informal) Under appropriate regularity assumptions, there is pessimistic policy optimization algorithm that, when trained in the MDP $M$ with the reward model $r_{\widetilde{\theta}_{\text{MLE}_\tau}}$ and data covariance estimate $\widetilde{\Sigma}_{\mathcal{D}_\tau}$ outputs a policy $\widetilde{\pi}_{PE}$ achieving expected rewards that are worse than those of the optimal policy by at most*

$$O\left(\sqrt{\frac{d}{n}} + \frac{(d \log(1/\delta))^{1/4}}{\sqrt{\epsilon n}}\right)$$

Again in the typical setting where $\epsilon$ is constant and $\delta$ is inverse polynomial in $n$, these results match the non-private algorithm of Zhu et al. (2023) up to logarithmic factors.

## 2 PRELIMINARIES

**Notation.** We use the notation $[K] = \{1, \ldots, K\}$. We write $\mathcal{N}(\mu, \sigma^2)^d$ to denote the distribution of random vector whose entries are independent Gaussian random variables with mean $\mu$ and variance $\sigma^2$. We use $\|\cdot\|_2$ to denote the standard $\ell_2$-norm on $\mathbb{R}^d$. For a positive semidefinite matrix $M \in \mathbb{R}^{d \times d}$ we define the semi-norm $\|v\|_M = \sqrt{v^\top M v}$ for any vector $v \in \mathbb{R}^d$. For a pair of matrices $A$ and $B$ we write $A \succcurlyeq B$ if and only if $A - B$ is positive semidefinite.

**Reinforcement learning** A finite-horizon Markov Decision Process (MDP) with horizon $H$ is represented by a tuple $(S, A, \{r_h\}_{h=1}^H, \{T_h\}_{h=1}^h, \rho_0)$. Here, $S$ represents the state space, $A$ represents the action space, and $\rho$ represents the initial state distribution. For each $h \in [H]$ there is a reward function $r_h : S \times A \to \mathbb{R}$ assigning a real-valued reward to each state-action pair, and a transition function $T_h : S \times H \to \Delta(S)$ taking a state-action pair to a distribution over states.

A deterministic policy $\pi = \{\pi_h\}_{h=1}^H$ is a collection of functions $\pi_h : S \to A$ giving an action $a$ to be taken in state $s$. A policy $\pi$ in an MDP $M$ induces a distribution over sequences of states and actions. In particular, first $s_1 \sim \rho_0$ and $a_1 = \pi_1(s_1)$, and then subsequently $s_h \sim T(s_{h-1}, a_{h-1})$ and $a_h = \pi_h(s_h)$ for each $h \in [H]$. The value function $V^\pi : S \to \mathbb{R}$ for the policy $\pi$ is then the expected cumulative rewards obtained when starting in state $s$,

$$V^\pi(s) = \mathbb{E}_{a_h = \pi_h(s_h)} \left[ \sum_{h=1}^H r(s_h, a_h) | s_1 = s \right].$$

We further define the occupancy measure $\rho_\pi$ of a policy $\pi$ to be the probability distribution over state-action pairs encountered when utilizing the policy $\pi$ in the MDP $M$,

$$\rho_\pi(s, a) = \mathop{\mathbb{P}}_{\substack{s_1 \sim \rho_0 \\ s_h \sim T(s_{h-1}, a_{h-1}), a_h = \pi_h(s_h)}} [s_h = s, a_h = a].$$

We use $\pi^* = \arg\max_\pi V^\pi$ to denote the optimal policy i.e. the policy that maximizes the expected cumulative rewards. The objective in reinforcement learning is to learn a policy $\hat{\pi}$ that obtains rewards that are close to those obtained by the optimal policy $\pi^*$. Formally, we define the suboptimality of a policy $\hat{\pi}$ by $\mathrm{SubOpt}(\hat{\pi}) = \mathbb{E}_{s \sim \rho_0}[V^{\pi^*}(s) - V^{\hat{\pi}}(s)]$. The setting where the horizon $H = 1$ is referred to as the contextual bandit setting. In particular, in this setting there are no transitions, and the state $s$ is always sampled from the fixed initial state distribution $\rho_0$. This is the setting that most accurately models RLHF as it is typically applied to language models.

**Reinforcement learning from human feedback** In reinforcement learning from human feedback the humans provide preference rankings over actions. Given a state $s$ and $K$ possible actions $(a_1, \ldots a_K)$, the ranking over the actions is given by a permutation $\sigma : [K] \to [K]$ that ranks the actions from the most preferred $a_\sigma(1)$, to the least preferred $a_\sigma(K)$. In RLHF these preference rankings are assumed to arise as samples from the Plackett-Luce model.

$$\mathbb{P}(\sigma | s, a_0, a_1, \ldots a_K) = \prod_{k=1}^K \frac{\exp(r^*(s, a_{\sigma(k)}))}{\sum_{j=m}^K \exp(r^*(s, a_{\sigma(j)}))}.$$

where $r^*(s, a)$ is a ground-truth reward function corresponding to underlying human preferences. The input to RLHF is then a data-set of human preference rankings $\mathcal{D} = \{(s^i, a_1^i, \ldots, a_k^i, \sigma^i)\}_{i=1}^n$, where the state $s^i$ and tuple of actions $a_1^i, \ldots, a_K^i$ can be arbitrary, but the preference ranking $\sigma^i$ is assumed to be sampled from the Plackett-Luce model.

Throughout the paper, we make the following assumption regarding the parameterization of the reward function $r^*$, which is the same as that made in prior work (Zhu et al., 2023).

**Assumption 2.1.** The reward function comes from a class of linear functions $r_\theta(s, a) = \langle \theta), \phi(s, a) \rangle$ with a known feature map $\phi : S \times A \to \mathbb{R}^d$ satisfying $\|\phi(s, a)\|_2 \leq L$ for all $(s, a)$. Further, we assume that the true parameter $\theta^*$ for the reward satisfies $\theta^* \in \Theta_B = \{\theta \mid \|\theta\|_2 \leq B\}$.

We denote ground-truth linear parameter vector $\theta^*$, so that $r^*(s, a) = r_{\theta^*}(s, a)$. In reinforcement learning from human feedback one first uses the dataset $\mathcal{D}$ to learn an estimated reward parameter $\hat{\theta}$, and then trains a policy $\hat{\pi}$ in the MDP $M$ using the learned reward $r_{\hat{\theta}}$. Critically, the objective is to obtain good performance relative to the ground-truth rewards $r_{\theta^*}$, despite training with an estimated reward function $r_{\hat{\theta}}$.

## 2.1 DIFFERENTIAL PRIVACY

Our results are stated in terms of the rigorous notion of *differential privacy*. Let $\mathcal{D}$ be a dataset containing $n$ items. In our case each item is a tuple $(s, a_1, \ldots, a_K, \sigma)$ representing human preference rankings. For another dataset $\mathcal{D}'$ we use the notation $\|\mathcal{D} - \mathcal{D}'\|_1 = 1$ to indicate that $\mathcal{D}$ and $\mathcal{D}'$ differ in exactly one item, and are otherwise identical. The formal definition of differential privacy is then,

**Definition 2.2.** $((\epsilon, \delta)$-differential privacy (Dwork and Roth, 2014)) A randomized algorithm $\mathcal{A}$ is $(\epsilon, \delta)$-differentially private if for all $\mathcal{O} \subseteq \text{Range}(\mathcal{A})$ such that $\|\mathcal{D} - \mathcal{D}'\|_1 \leq 1$:

$$\mathbb{P}[\mathcal{A}(\mathcal{D}) \in \mathcal{O}] \leq e^\epsilon \, \mathbb{P}[\mathcal{A}(\mathcal{D}') \in \mathcal{O}] + \delta \tag{1}$$

where the probability space is over the coin flips of the mechanism $\mathcal{A}$. When $\delta = 0$, we say that $\mathcal{A}$ satisfies $\epsilon$-differential privacy.

Intuitively, differential privacy ensures that if one of the items in $\mathcal{D}$ contains private data for some person, even if all the other items in $\mathcal{D}$ are revealed, the output of the algorithm $\mathcal{A}$ leaks a negligible amount of information about the user. In particular, the distribution of the output is approximately equal to what it would be if that user's item were not present at all.

## 3 RELATED WORK

**Learning from Ranking in Bandits and Reinforcement Learning:** The most closely related work is the paper of Zhu et al. (2023), which recently gave minimax optimal bounds for the suboptimality of policies trained via RLHF when the rewards are assumed to be linearly parametrized. We consider the same setting in our paper, but additionally achieve differential privacy for RLHF, while asymptotically maintaining the same bounds on the suboptimality of the learned policy.

**Privacy in Bandits and Reinforcement Learning:** Differential privacy has been explored in linear contextual bandits (Shariff and Sheffet, 2018; Neel and Roth, 2018; Huang et al., 2023), in stochastic bandits with a central trust model[1] (Mishra and Thakurta, 2015; Tossou and Dimitrakakis, 2016; Sajed and Sheffet, 2019; Azize and Basu, 2022; Charisopoulos et al., 2023), with the local model of trust (Kasiviswanathan et al., 2011; Tenenbaum et al., 2021; Chowdhury and Zhou, 2023), in adversarial bandits (Tossou and Dimitrakakis, 2017), and in tabular MDPs Vietri et al. (2020). Wang and Hegde (2019) uses reproducing kernel Hilbert space norm-bounded noise to ensure private value function approximation with respect to the number of states queried. The notion of joint differential privacy in tabular MDPs was later extended to the linear MDP setting where the transitions and the reward functions parameterized by linear functions (Luyo et al., 2021; Ngo et al., 2022). Garcelon et al. (2021) provides a lower bound for regret minimization in finite-horizon MDPs with local differential privacy (LDP) guarantees. However, in all of the aforementioned settings, the rewards are assumed to be part of the private input, and do not need to be learned from data as is necessary in the setting we consider.

## 4 PRIVATE RLHF FOR CONTEXTUAL BANDITS

In this section we give our main results for private RLHF in the contextual bandit setting. For clarity of presentation we begin with the case of pairwise comparisons (i.e. $K = 2$ in the Plackett-Luce model). We then describe how to extend these results to general $K$. The contextual bandit setting corresponds most closely to the current approach to aligning language models with human preferences. In particular, given a prompt $s$ multiple possible responses $a^i$ are sampled from the model. Human raters then give a preference ranking over the responses. This dataset of preference rankings over responses is then used as the dataset for training reward models to be used subsequently to tune the model via RL.

### 4.1 PAIRWISE COMPARISONS

In this setting the dataset $\mathcal{D}$ consists of $n$ tuples $(s^i, a_0^i, a_1^i, y^i)$ where $y^i \in \{0, 1\}$ is an indicator variable with $y^i = 0$ if the human rater preferred $a_0^i$ in state $s$ and $y^i = 1$ if $a_1^i$ was preferred. Given

---

[1]In the central model of trust the users are trust a central database curator who has access the raw user data (Dwork and Roth, 2014).

a true reward parameter vector $\theta^*$, the Plackett-Luce model for $K = 2$ reduces to the Bradley-Terry-Luce model,

$$\mathbb{P}[y = l \mid s, a_0, a_1] = \frac{\exp(r_{\theta^*}(s, a_l))}{\exp(r_{\theta^*}(s, a_0)) + \exp(r_{\theta^*}(s, a_1))}.$$

In this case, the log-likelihood of a parameter vector $\theta$ is given by,

$$\ell_{\mathcal{D}}(\theta) = -\frac{1}{n} \sum_{i=1}^n \log\Bigg( \mathbf{1}[y_i = 1] \cdot \frac{1}{1 + \exp(-\langle \theta, \phi(s^i, a_1^i) - \phi(s^i, a_0^i)\rangle)}$$
$$+ \mathbf{1}[y_i = 0] \cdot \left( 1 - \frac{1}{1 + \exp(-\langle \theta, \phi(s^i, a_1^i) - \phi(s^i, a_0^i)\rangle)} \right) \Bigg)$$

Furthermore, for pairwise comparisons we define the data covariance matrix by $\Sigma_{\mathcal{D}} = \frac{1}{n} \sum_{i=1}^n \left( \phi(s^i, a_1^i) - \phi(s^i, a_0^i) \right) \left( \phi(s^i, a_1^i) - \phi(s^i, a_0^i) \right)^\top$. In order to privately estimate the rewards we utilize a version of objective-perturbed MLE Algorithm 1, which was shown to achieve $(\epsilon, \delta)$-differential privacy in Bassily et al. (2019a) with the bulk of the analysis coming from a theorem of Kifer et al. (2012). While the privacy analysis of Kifer et al. (2012) applies quite generally, achieving tight error bounds on the distance of $\widetilde{\theta}_{\mathrm{MLE}}$ from the unperturbed MLE $\hat{\theta}_{\mathrm{MLE}} = \arg\min_{\theta \in \Theta_B} \ell_{\mathcal{D}}(\theta)$ is more complex. For general convex MLE, usually one requires strong convexity of the loss to achieve tight error bounds on the $\ell_2$-distance $\|\widetilde{\theta}_{\mathrm{MLE}} - \hat{\theta}_{\mathrm{MLE}}\|_2$. In the RLHF setting that we consider, we instead have strong convexity with respect to the dataset-dependent seminorm $\|\cdot\|_{\Sigma_{\mathcal{D}}}$. Further, in order for pessimistic policy optimization to succeed we must bound the error in terms of the noise-perturbed dataset dependent norm $\|\cdot\|_{\widetilde{\Sigma}_{\mathcal{D}} + \lambda I}$ for some $\lambda > 0$.

This is a significant difference, because the noise perturbation added in Algorithm 1 in order to achieve differential privacy is from a standard, spherical Gaussian. In particular, it turns out the error introduced by adding such noise will scale with the norm of a spherical Gaussian under $\|\cdot\|_{(\Sigma_{\mathcal{D}} + \lambda I)^{-1}}$, which may be much larger than the standard $\ell_2$-norm. Thus, a more delicate analysis is required which trades-off the perturbations need for privacy (which must be standard Gaussians) versus the norm which is actually useful in measuring the error of the MLE for the RLHF setting.

---

**Algorithm 1** Private MLE for $\ell_{\mathcal{D}}$

---

**Input:** Dataset $\mathcal{D}$, privacy parameters $\epsilon \le 1$, $\delta \le \frac{1}{n^2}$, optimization accuracy parameter $0 < \beta < \frac{1}{n}$, failure probability $\eta$.

1: Sample $b \sim \mathcal{N}(0, \sigma^2)^d$, for $\sigma^2 = \frac{40 L^2 \log(\frac{1}{\delta})}{\epsilon^2}$

2: Sample $w \sim \mathcal{N}(0, \nu^2)^d$, for $\nu^2 = \frac{40 \beta \log(\frac{1}{\delta})}{\alpha \epsilon^2}$.

3: Set $\alpha = 2 C \gamma \frac{\sqrt{d \log(1/\delta) \log(1/\eta)}}{\epsilon n}$.

4: Define

$$\widetilde{\ell}_{\mathcal{D}}(\theta) = \ell_{\mathcal{D}}(\theta) + \alpha \|\theta\|_2^2 + \frac{\langle b, \theta \rangle}{n}$$

5: Compute an approximate solution $\hat{\theta}$ satisfying

$$\widetilde{\ell}_{\mathcal{D}}(\hat{\theta}) - \min_{\theta \in \Theta_B} \widetilde{\ell}_{\mathcal{D}}(\theta) < \beta$$

6: **return** $\widetilde{\theta}_{\mathrm{MLE}} = \hat{\theta} + w$

---

Privacy for the estimated covariance matrix $\widetilde{\Sigma}_{\mathcal{D}}$ follows from a straightforward application of the standard Gaussian mechanism.

---

**Algorithm 2** Private $\Sigma_{\mathcal{D}}$

---

**Input:** Dataset $\mathcal{D} = \{(s^i, a_0^i, a_1^i, y^i)\}_{i=1}^n$, privacy parameters $\epsilon \leq 1, \delta \leq \frac{1}{n^2}$.
  1: Compute the data covariance:

$$\Sigma_{\mathcal{D}} = \frac{1}{n} \sum_{i=1}^n \left(\phi(s^i, a_1^i) - \phi(s^i, a_0^i)\right) \left(\phi(s^i, a_1^i) - \phi(s^i, a_0^i)\right)^\top$$

  2: Sample: $G \sim \mathcal{N}(0, \sigma^2)^{d \times d}$,  for $\sigma^2 = \frac{64 \log(\frac{1}{\delta}) L^4}{\epsilon^2 n^2}$
  3: **return** $\widetilde{\Sigma}_{\mathcal{D}} = \Sigma_{\mathcal{D}} + G$.

---

We can now state our main theorem regarding privacy of the reward parameters $\widetilde{\theta}_{\mathrm{MLE}}$ and the data covariance $\widetilde{\Sigma}_{\mathcal{D}}$.

**Theorem 4.1.** *Let $\epsilon, \delta > 0$, and $\widetilde{\theta}_{\mathrm{MLE}}$ be the output of Algorithm 1 and $\widetilde{\Sigma}_{\mathcal{D}}$ the output of Algorithm 2. Then the pair $(\widetilde{\theta}_{\mathrm{MLE}}, \widetilde{\Sigma}_{\mathcal{D}})$ satisfies $(\epsilon, \delta)$-differential privacy.*

The proof appears in Section A.3. Note that while the theorem statement is straightforward, the key is to accurately balance the privacy achieved against the need for accuracy of the perturbed estimates of $\widetilde{\Sigma}_{\mathcal{D}}$ and $\widetilde{\theta}_{\mathrm{MLE}}$.

To state the pessimistic policy optimization algorithm that will be applied to the private outputs $\widetilde{\theta}_{\mathrm{MLE}}$ and $\widetilde{\Sigma}_{\mathcal{D}}$ we define the confidence set of parameters

$$\Theta(\widetilde{\theta}_{\mathrm{MLE}}, \lambda) = \left\{\theta \in \Theta_B \mid \|\widetilde{\theta}_{\mathrm{MLE}} - \theta\|_{\widetilde{\Sigma}_{\mathcal{D}} + \lambda I} \leq F(n, d, \eta, \epsilon, \delta)\right\} \tag{2}$$

where,

$$F(n, d, \eta, \epsilon, \delta) = O\left(\sqrt{\frac{d}{n}} + \frac{(d \log(1/\eta) \log(1/\delta))^{1/4}}{\sqrt{\epsilon n}}\right). \tag{3}$$

We also set $\lambda$ once and for all as

$$\lambda = C \cdot \frac{\sqrt{d \log(1/\eta) \log(1/\delta)}}{\epsilon n}$$

where the constant $C$ is the one provided by Lemma A.9. Algorithm 3 gives the pessimistic policy optimization algorithm that we apply to the learned rewards and data covariance. Note that the algorithm takes the perturbed reward parameter $\widetilde{\theta}_{\mathrm{MLE}}$ and covariance $\widetilde{\Sigma}_{\mathcal{D}}$ as inputs, but does not access the private dataset $\mathcal{D}$ at all. Thus by standard post-processing, the output of Algorithm 3 also satisfies $(\epsilon, \delta)$-differential privacy.

---

**Algorithm 3** Pessimistic policy optimization

---

**Input:** Error tolerance $\eta$, reward parameters $\widetilde{\theta}_{\mathrm{MLE}}$, perturbed data covariance $\widetilde{\Sigma}_{\mathcal{D}}$, confidence set $\Theta(\widetilde{\theta}_{\mathrm{MLE}}, \lambda)$, reference vector $v \in \mathbb{R}^d$, and state distribution $\rho$.
  1: Set $\hat{J}(\pi) = \min_{\theta \in \Theta(\widetilde{\theta}_{\mathrm{MLE}}, \lambda)} \mathbb{E}_{s \sim \rho}[\langle \theta, \phi(s, \pi(s)) - v\rangle]$.
  2: **return** $\hat{\pi}_{\mathrm{PE}} = \arg\max_\pi \hat{J}(\pi)$.

---

**Theorem 4.2.** *Let $\hat{\pi}_{PE}$ be the output of Algorithm 3, and $F(n, d, \eta, \epsilon, \delta)$ be as in (3). Then with probability at least $1 - \eta$,*

$$\mathrm{SubOpt}(\hat{\pi}_{PE}) \leq F(n, d, \eta, \epsilon, \delta) \|(\Sigma_{\mathcal{D}} + \lambda I)^{-1} (\mathbb{E}_{s \sim \rho}[\phi(s, \pi^*(s)) - v])\|_2$$

*where the $O(\cdot)$ hides factors depending only on $L$ and $B$. In particular, when $\epsilon$ is constant and $\delta$ is inverse polynomial in $n$,*

$$\mathrm{SubOpt}(\hat{\pi}_{PE}) \leq \widetilde{O}\left(\sqrt{\frac{d}{n}}\right) \|(\Sigma_{\mathcal{D}} + \lambda I)^{-1} (\mathbb{E}_{s \sim \rho}[\phi(s, \pi^*(s)) - v])\|_2.$$

The proof appears in Section A.5. The factor

$$\|(\Sigma_{\mathcal{D}} + \lambda I)^{-1} \left(\mathbb{E}_{s \sim \rho}[\phi(s, \pi^*(s)) - v]\right)\|_2$$

is known as the *single concentratability coefficient*, and is a measure of how well the offline dataset covers the average feature vector $\mathbb{E}_{s \sim \rho}[\phi(s, \pi^*(s))]$. The same factor appears in Zhu et al. (2023) and other related work on offline reinforcement learning. In particular, it is standard practice to assume that the single concentratability coefficient is bounded by a constant independent of $d$ and $n$.

The vector $v$ is free to be chosen by the algorithm designer, and can make a significant difference in the magnitude of the bound. See Zhu et al. (2023) for an example of a simple multiarmed bandit setting where $\mathbb{E}_{s \sim \rho}[\phi(s, \pi^*(s))]$ is in the null space of $\Sigma_{\mathcal{D}}$, and hence

$$\|(\Sigma_{\mathcal{D}} + \lambda I)^{-1} \left(\mathbb{E}_{s \sim \rho}[\phi(s, \pi^*(s))]\right)\|_2 \to \infty$$

as $\lambda \to 0$. However, for the same MDP there exists a $v$ such that

$$\|(\Sigma_{\mathcal{D}} + \lambda I)^{-1} \left(\mathbb{E}_{s \sim \rho}[\phi(s, \pi^*(s)) - v]\right)\|_2 \leq 1.$$

It is also critical to note that the error bound is given in terms of $(\Sigma_{\mathcal{D}} + \lambda I)^{-1}$ and not $\left(\widetilde{\Sigma}_{\mathcal{D}} + \lambda I\right)^{-1}$. That is, even though the pessimistic policy optimization algorithm only has access to $\widetilde{\Sigma}_{\mathcal{D}}$ the error depends on the *true value* of the single concentratability coefficient determined by $\Sigma_{\mathcal{D}}$, and thus makes our results directly comparable to the non-private algorithm. This introduces additional subtleties in our proof, which do not appear in the non-private case where the pessimistic policy algorithm has access to the unperturbed $\Sigma_{\mathcal{D}}$.

### 4.2 $K$-WISE COMPARISONS

For the case of $K$-wise comparisons the dataset $\mathcal{D}_K$ consists of $n$ tuples of the form $(s^i, a_1^i, \ldots, a_K^i, \sigma)$, where $\sigma$ is a permutation on $K$ elements representing a human preference ranking. The log likelihood for the Plackett-Luce model with general $K$ takes the form,

$$\ell_{\mathcal{D}_K}(\theta) = -\frac{1}{n} \sum_{i=1}^{n} \sum_{j=1}^{K} \log \left( \frac{\exp \left( \left\langle \theta, \phi(s^i, a_{\sigma_i(j)}^i) \right\rangle \right)}{\sum_{k=j}^{K} \exp \left( \left\langle \theta, \phi(s^i, a_{\sigma_i(k)}^i) \right\rangle \right)} \right).$$

In this case the data covariance matrix is given by

$$\Sigma_{\mathcal{D}_K} = \frac{2}{nK(K-1)} \sum_{i=1}^{n} \sum_{j=1}^{K} \sum_{j=k+1}^{K} \left( (\phi(s^i, a_j^i) - \phi(s^i, a_k^i))(\phi(s^i, a_j^i) - \phi(s^i, a_k^i))^\top \right)$$

The main subtlety in extending our main privacy result Theorem 4.1 to the setting of $K$-wise comparisons relates to the assumptions required for objective-perturbed MLE as in Algorithm 1 to maintain privacy. In particular, the loss takes the form of a sum of $n$ terms

$$\ell_{\mathcal{D}_K}(\theta) = \sum_{i=1}^{n} \ell_{\mathcal{D}_K}^i(\theta),$$

where $\ell_{\mathcal{D}_K}^i$ is determined by the tuple $(s^i, a_1^i, \ldots a_K^i, \sigma_i) \in \mathcal{D}_K$. By linearity, the Hessian is given by

$$\nabla^2 \ell_{\mathcal{D}_K}(\theta) = \sum_{i=1}^{n} \nabla^2 \ell_{\mathcal{D}_K}^i(\theta).$$

As stated, the original privacy theorem of Kifer et al. (2012) only applies under the assumption that each such term $\nabla^2 \ell_{\mathcal{D}_K}^i(\theta)$ has rank one. Unfortunately, this is false for our case, as $\nabla^2 \ell_{\mathcal{D}_K}^i(\theta)$ may actually have rank as large as $K^3$. Luckily, as shown in Bassily et al. (2019b), the results of Iyengar et al. (2019) can be applied to allow for constant rank for the individual Hessians $\nabla^2 \ell_{\mathcal{D}_K}^i(\theta)$ to achieve differential privacy. In particular, we show that we can adjust $\alpha$ by a constant factor depending on $K$ in order to satisfy the appropriate assumptions to achieve privacy. Further, privacy for $\widetilde{\Sigma}_{\mathcal{D}_K}$ output by Algorithm 2 applied to the dataset $\mathcal{D}_K$ follows again from the standard Gaussian mechanism. Thus, altogether we can prove our main privacy theorem.

**Theorem 4.3.** *Let $\epsilon, \delta > 0$, and $\widetilde{\theta}_{\mathrm{MLE}_K}$ be the output of Algorithm 1 (with parameters modified by a constant factor) and $\widetilde{\Sigma}_{\mathcal{D}_K}$ the output of Algorithm 2, when applied to the dataset $\mathcal{D}_K$. Then the pair $(\widetilde{\theta}_{\mathrm{MLE}_K}, \widetilde{\Sigma}_{\mathcal{D}_K})$ satisfies $(\epsilon, \delta)$-differential privacy.*

The proof appears in Section B.3. For the pessimistic policy optimization algorithm applied to $K$-wise comparisons, we define a similar confidence set

$$\Theta_K(\widetilde{\theta}_{\mathrm{MLE}_K}, \lambda) = \left\{ \theta \in \Theta_B \mid \|\widetilde{\theta}_{\mathrm{MLE}} - \theta\|_{\widetilde{\Sigma}_{\mathcal{D}} + \lambda I} F(n, d, \eta, \epsilon, \delta) \right\} \tag{4}$$

where $F(n, d, \eta, \epsilon, \delta)$ is given by (3). Finally, our main theorem on the performance of pessimistic policy optimization follows by running Algorithm 3 on $\mathcal{D}_K$ with confidence set $\Theta(\widetilde{\theta}_{\mathrm{MLE}_K}, \lambda)$.

**Theorem 4.4.** *Let $\hat{\pi}_{PE}$ be the output of Algorithm 3 when run with input $\widetilde{\theta}_{\mathrm{MLE}_K}, \widetilde{\Sigma}_{\mathcal{D}_K}$, and confidence set $\Theta_K(\widetilde{\theta}_{\mathrm{MLE}_K}, \lambda)$. Let $F(n, d, \eta, \epsilon, \delta)$ be as in (3). Then with probability at least $1 - \eta$,*

$$\mathrm{SubOpt}(\hat{\pi}_{PE}) \leq F(n, d, \eta, \epsilon, \delta)\|(\Sigma_{\mathcal{D}} + \lambda I)^{-1} (\mathbb{E}_{s \sim \rho}[\phi(s, \pi^*(s)) - v])\|_2$$

*where the $O(\cdot)$ hides factors depending only on $L$, $B$, and $K$. In particular, when $\epsilon$ is constant and $\delta$ is inverse polynomial in $n$,*

$$\mathrm{SubOpt}(\hat{\pi}_{PE}) \leq \widetilde{O}\left(\sqrt{\frac{d}{n}}\right) \|(\Sigma_{\mathcal{D}} + \lambda I)^{-1} (\mathbb{E}_{s \sim \rho}[\phi(s, \pi^*(s)) - v])\|_2.$$

The proof appears in Section B.

## 5 PRIVATE RLHF FOR GENERAL MDPS

In this section we extend our results to private RLHF in finite-horizon MDPs. In this case we start with a set of trajectories i.e. length $H$ sequences of state-action pairs $\tau^i = ((s_1^i, a_1^i), (s_2^i, a_2^i), \ldots (s_H^i, a_H^i))$. Then human ratings of pairs of trajectories are made to produce a dataset $\mathcal{D}_\tau = \{\tau_0^i, \tau_1^i, y^i\}_{i=1}^n$, where $y_i = l$ for $l \in \{0, 1\}$ implies that the human preferred trajectory $\tau_l^i$. Here $\tau_0^i$ and $\tau_1^i$ both start with the same state. Once again we assume that given a ground-truth parameter vector $\theta^*$, the human preference ratings follow a Bradley-Terry-Luce model of the form,

$$\mathbb{P}[y = 1 \mid s, \tau_0, \tau_1] = \frac{\exp\left(\sum_{h=1}^H r_{\theta^*}(s_{h1}, a_{h1})\right)}{\sum_{j \in \{0,1\}} \exp\left(\sum_{h=1}^H r_{\theta^*}(s_{hj}, a_{hj})\right)}$$

where above

$$\tau_0 = ((s_{10}, a_{10}), (s_{20}, a_{20}), \ldots (s_{H0}, a_{H0})) \text{ and } \tau_1 = ((s_{11}, a_{11}), (s_{21}, a_{21}), \ldots (s_{H1}, a_{H1})).$$

In this setting the log-likelihood is given by

$$\ell_{\mathcal{D}_\tau}(\theta) = -\frac{1}{n} \sum_{i=1}^n \log \left( \frac{\mathbf{1}[y_i = 1] \cdot \exp\left(\sum_{h=1}^H r_{\theta^*}(s_{h1}^i, a_{h1}^i)\right)}{\sum_{j \in \{0,1\}} \exp\left(\sum_{h=1}^H r_{\theta^*}(s_{hj}^i, a_{hj}^i)\right)} \right.$$
$$\left. + \frac{\mathbf{1}[y_i = 0] \cdot \exp\left(\sum_{h=1}^H r_{\theta^*}(s_{h0}, a_{h0})\right)}{\sum_{j \in \{0,1\}} \exp\left(\sum_{h=1}^H r_{\theta^*}(s_{hj}, a_{hj})\right)} \right)$$

The relevant data covariance matrix is

$$\Sigma_{\mathcal{D}_\tau} = \frac{1}{n} \sum_{i=1}^n \left( \sum_{h=1}^H \left(\phi(s_{h0}^i, a_{h0}^i) - \phi(s_{h1}^i, a_{h1}^i)\right) \left(\phi(s_{h0}^i, a_{h0}^i) - \phi(s_{h1}^i, a_{h1}^i)\right)^\top \right).$$

As in the contextual bandit case, we run Algorithm 1 with the dataset of trajectories $\mathcal{D}_\tau$ to produce a parameter estimate $\widetilde{\theta}_{\mathrm{MLE}_\tau}$. Further, we modify Algorithm 2 to use the trajectory covariance matrix $\Sigma_{\mathcal{D}_\tau}$ given above, resulting in private trajectory covariance output $\widetilde{\Sigma}_{\mathcal{D}_\tau}$. We then have the following theorem.

**Theorem 5.1.** *Let $\epsilon, \delta > 0$, and $\widetilde{\theta}_{\mathrm{MLE}_\tau}$ be the output of Algorithm 1 and $\widetilde{\Sigma}_{\mathcal{D}_\tau}$ the output of Algorithm 2 when run on the trajectory dataset $\mathcal{D}$. Then the pair $(\widetilde{\theta}_{\mathrm{MLE}_\tau}, \widetilde{\Sigma}_{\mathcal{D}_\tau})$ satisfies $(\epsilon, \delta)$-differential privacy.*

The proof appears in C.3.

In order to utilize Algorithm 3 for the general MDP setting, one needs to consider the distribution $\rho_\pi$ on states induced by the utilization of the policy $\pi$ in the MDP $M$. In this case the pessimistic policy loss function in Algorithm 3 becomes

$$\hat{J}(\pi) = \min_{\theta \in \Theta(\widetilde{\theta}_{\mathrm{MLE}_\tau}, \lambda)} \mathbb{E}_{s \sim \rho_\pi}[r_{\widetilde{\theta}_{\mathrm{MLE}_\tau}}(s, \pi(s))].$$

Slightly abusing notation, we will refer to the use of this loss function as running Algorithm 3 with input $\rho = \rho_\pi$.

**Theorem 5.2.** *Let $\widetilde{\theta}_{\mathrm{MLE}_\tau}$ and $\widetilde{\Sigma}_{\mathcal{D}_\tau}$ be as in Theorem 5.1. Let $\hat{\pi}_{PE}$ be the output of Algorithm 3 when run with $\rho = \rho_\pi$, and $F(n, d, \eta, \epsilon, \delta)$ as in (3). Then with probability at least $1 - \eta$,*

$$\mathrm{SubOpt}(\hat{\pi}_{PE}) \leq F(n, d, \eta, \epsilon, \delta) \cdot \|(\Sigma_\mathcal{D} + \lambda I)^{-1} (\mathbb{E}_{s \sim \rho_\pi}[\phi(s, \pi^*(s)) - v])\|_2$$

*where the $O(\cdot)$ hides factors depending only on $L, H$, and $B$. In particular, when $\epsilon$ is constant and $\delta$ is inverse polynomial in $n$,*

$$\mathrm{SubOpt}(\hat{\pi}_{PE}) \leq \widetilde{O}\left(\sqrt{\frac{d}{n}}\right) \cdot \|(\Sigma_\mathcal{D} + \lambda I)^{-1} (\mathbb{E}_{s \sim \rho_\pi}[\phi(s, \pi^*(s)) - v])\|_2.$$

The proof appears in Section C.

## 6 CONCLUSION AND OPEN PROBLEMS

We have shown that it is possible to perform reinforcement learning from human feedback with minimax optimal rates and differential privacy when rewards are linearly parametrized. The setting of linear parametrization in a fixed feature space is often used as a theoretical model in order to give qualitative insight into real-world machine learning algorithms. We view our results as qualitatively suggesting that it may be possible to simultaneously align large language models using RLHF while simultaneously protecting the privacy of the humans whose preference rankings are used during training. The ability to provide rigorous privacy guarantees can provably prevent the types of leaks of personal data described in The New York Times (2024), where a personal email address was leaked by a popular chat bot built on a large language model. The problem of privacy leaks due to LLMs is likely to only grow more serious as these models are utilized more widely, and differential privacy can be an important part of the solution.

A natural avenue for future work is to see if these theoretical results can be extended beyond linear parameterization. For instance, it would be interesting to study the setting where the rewards $r$ lie in a general PAC-learnable function class, and then attempt to achieve statistical efficiency alongside differential privacy in such a setting.

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

# A PROOFS FOR CONTEXTUAL BANDITS WITH PAIRWISE COMPARISONS

## A.1 BASIC PROPERTIES OF $\ell_\mathcal{D}$ AND $\Sigma_\mathcal{D}$

We begin with the basic properties of $\ell_\mathcal{D}$ and $\Sigma_\mathcal{D}$ necessary for the analysis. Throughout we will use the notation $x_i = \phi(s^i, a_1^i) - \phi(s^i, a_0^i)$. With this notation the loss function $\ell_\mathcal{D}$ becomes

$$\ell_\mathcal{D}(\theta) = -\frac{1}{n} \sum_{i=1}^{n} \log \left( \left( \mathbf{1}[y_i = 1] \frac{1}{1 + \exp(-\langle \theta, x_i \rangle)} + \mathbf{1}[y_i = 0] \left( 1 - \frac{1}{1 + \exp(-\langle \theta, x_i \rangle)} \right) \right) \right) \tag{5}$$

The gradient and Hessian of $\ell_\mathcal{D}$ are given by the following formulas.

*Claim* A.1.

$$\nabla \ell_\mathcal{D}(\theta) = -\frac{1}{n} \sum_{i=1}^{n} \left( \mathbf{1}[y_i = 1] \frac{\exp(-\langle \theta, x_i \rangle)}{1 + \exp(-\langle \theta, x_i \rangle)} - \mathbf{1}[y_i = 0] \frac{1}{1 + \exp(-\langle \theta, x_i \rangle)} \right) x_i$$

*Claim* A.2.

$$\nabla^2 \ell_\mathcal{D}(\theta) = \frac{1}{n} \sum_{i=1}^{n} \frac{\exp(-\langle \theta, x_i \rangle)}{(1 + \exp(-\langle \theta, x_i \rangle))^2} x_i x_i^\top$$

*Proof.*

$$\nabla^2 \ell_\mathcal{D}(\theta) = \frac{1}{n} \sum_{i=1}^{n} \left( \mathbf{1}[y_i = 1] \frac{\exp(-\langle \theta, x_i \rangle)}{(1 + \exp(-\langle \theta, x_i \rangle))^2} + \mathbf{1}[y_i = 0] \frac{\exp(-\langle \theta, x_i \rangle)}{(1 + \exp(-\langle \theta, x_i \rangle))^2} \right) x_i x_i^\top$$

$$= \frac{1}{n} \sum_{i=1}^{n} \frac{\exp(-\langle \theta, x_i \rangle)}{(1 + \exp(-\langle \theta, x_i \rangle))^2} x_i x_i^\top$$

$\square$

These formulas lead directly to an upper bound on the norm of the gradient and the operator norm of the Hessian of $\ell_\mathcal{D}$.

**Lemma A.3.** *For all* $\theta$,

1. $\|\nabla \ell_\mathcal{D}(\theta)\|_2 \leq 2L$

2. $\|\nabla^2 \ell_\mathcal{D}(\theta)\|_{op} \leq 4L^2$

*Proof.* Observe first that $\|x_i\|_2 \leq 2L$ because $\|\phi(s,a)\| \leq L$. By Claim A.1, the gradient $\nabla \ell_\mathcal{D}(\theta)$ is the average of $n$ vectors each of length at most $2L$. Similarly by Claim A.2, $\nabla^2 \ell_\mathcal{D}(\theta)$ is the average of $n$ rank-one matrices, each of operator norm at most $\|x_i\|_2^2 \leq 4L^2$. $\square$

The proof Lemma 3.1 in Zhu et al. (2023) implies that for all $\theta \in \Theta_B$ and $v \in \mathbb{R}^d$

$$v^\top \nabla^2 \ell_\mathcal{D}(\theta) v \geq \gamma v^\top \Sigma_\mathcal{D} v = \gamma \|v\|_{\Sigma_\mathcal{D}}^2. \tag{6}$$

where $\gamma = 1/(2 + \exp(2LB) + \exp(-2LB))$. In particular, we have the following lemma,

**Lemma A.4.** $\ell_\mathcal{D}$ *is strongly convex on the set* $\Theta_B$ *with respect to the semi-norm* $\|\cdot\|_{\Sigma_\mathcal{D}}$. *That is, there exists a constant* $\gamma > 0$ *such that,*

$$\ell_\mathcal{D}(\theta + \Delta) - \ell_\mathcal{D}(\theta) - \langle \nabla \ell_\mathcal{D}(\theta), \theta \rangle \geq \frac{\gamma}{2} \|\Delta\|_{\Sigma_\mathcal{D}}^2 \tag{7}$$

*for all* $\theta \in \Theta_B$, *and* $\Delta$ *such that* $(\theta + \Delta) \in \Theta_B$.

We will need the following standard fact regarding optimizers of strongly convex functions over convex sets.

**Lemma A.5.** *Let* $\mathcal{C} \subseteq \mathbb{R}^d$ *be a convex set, let* $M \in \mathbb{R}^{d \times d}$ *be a positive semidefinite matrix, and let* $f : \mathbb{R}^d \to \mathbb{R}$ *be* $\gamma$-*strongly convex with respect to the seminorm* $\|\cdot\|_M$ *on* $\mathcal{C}$. *Let* $\hat{\theta}$ *be the minimum of* $f$ *in* $\mathcal{C}$. *Then* $f(\hat{\theta}) - f(\theta) \geq \frac{\gamma}{2} \|\hat{\theta} - \theta\|_M^2$ *for any point* $\theta \in \mathcal{C}$.

*Proof.* Follows from the second-order Taylor expansion of $f$ and the optimality conditions for optimization over a convex set. Then (6) implies the desired result. □

The following lemma allows us to quantify the effect of adding an $\ell_2$-norm regularizer to a function that is strongly convex with respect to a seminorm of the form $\|\cdot\|_M$.

**Lemma A.6.** *Let $M \in \mathbb{R}^{d \times d}$ be a positive semidefinite matrix. Suppose $f : \mathbb{R}^d \to \mathbb{R}$ is $\gamma$-strongly convex with respect to $\|\cdot\|_M$. Then the function $g(\theta) = f(\theta) + \frac{c}{2}\|\theta\|_2^2$ is $\gamma$-stongly convex with respect to $\|\cdot\|_{M+c/\gamma I}$.*

*Proof.*

$$\nabla^2 g(\theta) = \nabla^2 f(\theta) + cI \succeq \gamma \left( M + \frac{c}{\gamma} I \right)$$

□

### A.2 PRIVATE COVARIANCE

We obtain privacy for the feature covariance matrix via the Gaussian mechanism.

**Lemma A.7.** *Let $\sigma^2 = \frac{64 \log(\frac{1}{\delta})L^4}{\epsilon^2 n^2}$ and $G \sim \mathcal{N}(0, \sigma^2)^{d \times d}$. Then $\widetilde{\Sigma}_{\mathcal{D}} = \Sigma_{\mathcal{D}} + G$ is $(\epsilon/2, \delta/2)$-differentially private.*

*Proof.* For a dataset $\mathcal{D}'$ differing in one query $(s, a_0, a_1)$ from $\mathcal{D}$ we have

$$\|\Sigma_{\mathcal{D}} - \Sigma_{\mathcal{D}'}\|_2 \leq \frac{1}{n} \|(\phi(s, a_1) - \phi(s, a_0))(\phi(s, a_1) - \phi(s, a_0))^\top\|_2 = \frac{1}{n} \|\phi(s, a_1) - \phi(s, a_0)\|_2^2 \leq \frac{4L^2}{n}.$$

The standard analysis of the Gaussian mechanism (Dwork and Roth, 2014) then implies that $\widetilde{\Sigma}_{\mathcal{D}}$ is $(\epsilon/2, \delta/2)$-differentially private when setting $\sigma^2 = \frac{64 \log(\frac{1}{\delta})L^4}{\epsilon^2 n^2}$. □

The parameter estimation error is asymptotically the same when measuring with respect to the differentially private covariance matrix $\widetilde{\Sigma}_{\mathcal{D}}$.

**Lemma A.8.** *Let $z \in \mathbb{R}^d$. With probability at least $1 - \eta$,*

$$\|z\|_{\widetilde{\Sigma}_{\mathcal{D}} + \lambda I} < \sqrt{1 + O\left( \frac{\sqrt{\log(1/\delta) \log(1/\eta)}}{\epsilon^2 n^2 \lambda} \right)} \|z\|_{\Sigma_{\mathcal{D}} + \lambda I}$$

*Proof.* Since $\widetilde{\Sigma}_{\mathcal{D}} = \Sigma_{\mathcal{D}} + G$ for $G \sim \mathcal{N}(0, \sigma^2)^{d \times d}$,

$$\|z\|_{\widetilde{\Sigma}_{\mathcal{D}} + \lambda I}^2 = \|z\|_{\Sigma_{\mathcal{D}} + \lambda I}^2 + z^\top G z \tag{8}$$

Further $z^\top G z$ is a linear function of the independent $\mathcal{N}(0, \sigma^2)$ entries of $G$, and thus is distributed as a Gaussian with mean 0 and variance $\sigma^2 \|z\|_2^4$. Next note that since $\Sigma_{\mathcal{D}}$ is positive semidefinite,

$$\lambda \|z\|_2^2 = z^\top \lambda I z$$
$$\leq z^\top (\Sigma_{\mathcal{D}} + \lambda I) z = \|z\|_{\Sigma_{\mathcal{D}} + \lambda I}^2. \tag{9}$$

Thus by (9) and standard Gaussian concentration, with probability at least $1 - \eta$,

$$z^\top G z < \sqrt{\log\left(\frac{1}{\eta}\right)} \sigma \|z\|_2^2$$

$$\leq \sqrt{\log\left(\frac{1}{\eta}\right)} \frac{\sigma}{\lambda} \|z\|_{\Sigma_{\mathcal{D}} + \lambda I}^2$$

$$\leq O\left( \frac{\sqrt{\log(1/\delta) \log(1/\eta)}}{\epsilon^2 n^2 \lambda} \right) \|z\|_{\Sigma_{\mathcal{D}} + \lambda I}^2$$

Plugging into (8) and taking square roots yields the desired result. □

We next prove bounds relating $(\Sigma_{\mathcal{D}} + \lambda I)^{-1}$ to $(\widetilde{\Sigma}_{\mathcal{D}} + \lambda I)^{-1}$.

**Lemma A.9.** *There is a constant $C > 0$ such that for $\lambda \geq C\frac{\sqrt{d\log(1/\eta)\log(1/\delta)}}{\epsilon n}$ we have*

$$\|(\widetilde{\Sigma}_{\mathcal{D}} + \lambda I)^{-1/2}z\|_2 \leq \left\|\left(\Sigma_{\mathcal{D}} + \frac{\lambda}{2}I\right)^{-1/2}z\right\|_2$$

*Proof.* Note that $\widetilde{\Sigma}_{\mathcal{D}} = \Sigma_{\mathcal{D}} + G$ where $G \sim \mathcal{N}(0, \sigma^2)^{d \times d}$, for $\sigma^2 = \frac{64\log(\frac{1}{\delta})L^4}{\epsilon^2 n^2}$. Therefore by standard concentration bounds for the operator norm of a matrix with independent Gaussian entries Vershynin (2018) we have that with probability at least $1 - \eta$,

$$\|G\|_{\mathrm{op}} \leq C'\sigma(\sqrt{d} + \sqrt{\log(1/\eta)})$$
$$\leq C''\frac{\sqrt{d\log(1/\delta)\log(1/\eta)}}{\epsilon n}.$$

Next set $C = 2C''$, and let $\mu = \|G\|_{\mathrm{op}}$. Then, with probability at least $1 - \eta$,

$$\widetilde{\Sigma}_{\mathcal{D}} + \lambda I = \Sigma_{\mathcal{D}} + G + \lambda I \succcurlyeq \Sigma_{\mathcal{D}} + (\lambda - \mu)I = \Sigma_{\mathcal{D}} + \frac{\lambda}{2}I.$$

Therefore,

$$z^\top(\widetilde{\Sigma}_{\mathcal{D}} + \lambda I)^{-1}z \leq z^\top(\Sigma_{\mathcal{D}} + \frac{\lambda}{2}I)^{-1}z.$$

Taking square roots yields the desired result. $\qquad\square$

### A.3 PRIVACY OF OBJECTIVE-PERTURBED MLE

**Lemma A.10.** *Algorithm 1 satisfies $(\epsilon/2, \delta/2)$-differential privacy.*

*Proof.* For the chosen values of $\alpha, \sigma$, and $\nu$ given in Algorithm 1, the function $\ell_{\mathcal{D}}$ satsifies the assumptions of Theorem 5.6 of Bassily et al. (2019b) which is the full version of Bassily et al. (2019a). Further note that Theorem 5.6 of Bassily et al. (2019b) is just output perturbation applied to the objective perturbation from Theorem 2 in Kifer et al. (2012). $\qquad\square$

We now have all the ingredients necessary to prove our main result on differential privacy for the setting of contextual bandits with pairwise comparisons.

*Proof of Theorem 4.1.* $\widetilde{\theta}_{\mathrm{MLE}}$ is $(\epsilon/2, \delta/2)$-differentially private by Lemma A.10, and $\widetilde{\Sigma}_{\mathcal{D}}$ is $(\epsilon/2, \delta/2)$-differentially private by Lemma A.7. Thus, standard composition implies that the pair $(\widetilde{\theta}_{\mathrm{MLE}}, \widetilde{\Sigma}_{\mathcal{D}})$ is $(\epsilon, \delta)$-differentially private. $\qquad\square$

### A.4 APPROXIMATION ERROR OF OBJECTIVE-PERTURBED MLE

We now prove an upper bound on the distance between the output of Algorithm 1 and the true MLE solution.

**Lemma A.11.** *Let $\lambda = C\frac{\sqrt{d\log(1/\eta)\log(1/\delta)}}{\epsilon n}$, with probability at least $1 - \eta$,*

$$\|\hat{\theta}_{\mathrm{MLE}} - \widetilde{\theta}_{\mathrm{MLE}}\|_{\widetilde{\Sigma}_{\mathcal{D}}+\lambda I} \leq O\left(\frac{(d\log(1/\eta)\log(1/\delta))^{1/4}}{\sqrt{\epsilon n}}\right)$$

*where the $O(\cdot)$ hides factors depending only on $L$ and $B$.*

*Proof.* Let $\alpha, \sigma^2$, and $b$ be as in Algorithm 1. First, define the $\ell_2$-regularized and objective-perturbed loss functions as follows:

$$\ell_{\mathcal{D}}'(\theta) = \ell_{\mathcal{D}}(\theta) + \alpha\|\theta\|_2^2 \tag{10}$$

$$\widetilde{\ell}_{\mathcal{D}}(\theta) = \ell_{\mathcal{D}}(\theta) + \alpha\|\theta\|_2^2 + \frac{\langle b, \theta\rangle}{n} \tag{11}$$

Further let $\hat{\theta}_{\mathrm{MLE}} = \arg\min_{\theta\in\Theta_B}\ell_{\mathcal{D}}(\theta)$, $\theta' = \arg\min_{\theta\in\Theta_B}\ell_{\mathcal{D}}'(\theta)$, and $\hat{\theta} = \arg\min_{\theta\in\Theta_B}\widetilde{\ell}_{\mathcal{D}}(\theta)$.

**An upper bound for $\|\hat{\theta}_{\mathbf{MLE}} - \theta'\|$.** By Lemma A.4 and Lemma A.6 the loss $\ell'_{\mathcal{D}}(\theta)$ is $\gamma$-strongly convex with respect to $\|\cdot\|_{\Sigma_{\mathcal{D}} + \frac{\alpha}{\gamma}I}$. Thus, Lemma A.5 implies that

$$\ell'_{\mathcal{D}}(\hat{\theta}_{\mathrm{MLE}}) \geq \ell'_{\mathcal{D}}(\theta') + \frac{\gamma}{2}\|\hat{\theta}_{\mathrm{MLE}} - \theta'\|^2_{\Sigma_{\mathcal{D}} + \frac{\alpha}{\gamma}I}$$

$$\implies \ell_{\mathcal{D}}(\hat{\theta}_{\mathrm{MLE}}) + \alpha\|\hat{\theta}_{\mathrm{MLE}}\|^2_2 \geq \ell_{\mathcal{D}}(\theta') + \alpha\|\theta'\|^2_2 + \frac{\gamma}{2}\|\hat{\theta}_{\mathrm{MLE}} - \theta'\|^2_{\Sigma_{\mathcal{D}} + \frac{\alpha}{\gamma}I}$$

Observe that $\ell_{\mathcal{D}}(\hat{\theta}_{\mathrm{MLE}}) \leq \ell_{\mathcal{D}}(\theta')$ by optimality of $\hat{\theta}_{\mathrm{MLE}}$. Thus,

$$\alpha\|\hat{\theta}_{\mathrm{MLE}}\|^2_2 \geq \alpha\|\theta'\|^2_2 + \frac{\gamma}{2}\|\hat{\theta}_{\mathrm{MLE}} - \theta'\|^2_{\Sigma_{\mathcal{D}} + \frac{\alpha}{\gamma}I}$$

$$\geq \frac{\gamma}{2}\|\hat{\theta}_{\mathrm{MLE}} - \theta'\|^2_{\Sigma_{\mathcal{D}} + \frac{\alpha}{\gamma}I}$$

Rearranging and using the fact that $\|\hat{\theta}_{\mathrm{MLE}}\|_2 \leq B$ yields

$$\|\hat{\theta}_{\mathrm{MLE}} - \theta'\|_{\Sigma_{\mathcal{D}} + \frac{\alpha}{\gamma}I} \leq \sqrt{\frac{2\alpha B}{\gamma}} \tag{12}$$

**An upper bound for $\|\hat{\theta} - \theta'\|$.** Adding a linear term has no affect on strong convexity, thus by Lemma A.4 and Lemma A.6 the function $\widetilde{\ell}_{\mathcal{D}}(\theta)$ is $\gamma$-strongly convex with respect to $\|\cdot\|_{\Sigma_{\mathcal{D}} + \frac{\alpha}{\gamma}I}$. Again Lemma A.5 implies

$$\widetilde{\ell}_{\mathcal{D}}(\theta') \geq \widetilde{\ell}_{\mathcal{D}}(\hat{\theta}) + \frac{\gamma}{2}\|\hat{\theta} - \theta'\|^2_{\Sigma_{\mathcal{D}} + \frac{\alpha}{\gamma}I}$$

$$\implies \ell'_{\mathcal{D}}(\theta') + \frac{\langle b, \theta'\rangle}{n} \geq \ell'_{\mathcal{D}}(\hat{\theta}) + \frac{\langle b, \hat{\theta}\rangle}{n} + \frac{\gamma}{2}\|\hat{\theta} - \theta'\|^2_{\Sigma_{\mathcal{D}} + \frac{\alpha}{\gamma}I}$$

By the optimality of $\theta'$ for $\ell'_{\mathcal{D}}$, we have $\ell'_{\mathcal{D}}(\hat{\theta}_{\mathrm{MLE}}) \geq \ell'_{\mathcal{D}}(\theta')$. Hence,

$$\frac{\langle b, \theta'\rangle}{n} \geq \frac{\langle b, \hat{\theta}\rangle}{n} + \frac{\gamma}{2}\|\hat{\theta} - \theta'\|^2_{\Sigma_{\mathcal{D}} + \frac{\alpha}{\gamma}I}$$

$$\implies \langle b, \theta' - \hat{\theta}\rangle \geq \frac{n\gamma}{2}\|\hat{\theta} - \theta'\|^2_{\Sigma_{\mathcal{D}} + \frac{\alpha}{\gamma}I}.$$

Therefore by Cauchy-Schwarz,

$$\|b\|_{(\Sigma_{\mathcal{D}} + \frac{\alpha}{\gamma}I)^{-1}}\|\hat{\theta} - \theta'\|^2_{\Sigma_{\mathcal{D}} + \frac{\alpha}{\gamma}I} \geq \frac{n\gamma}{2}\|\hat{\theta} - \theta'\|^2_{\Sigma_{\mathcal{D}} + \frac{\alpha}{\gamma}I}.$$

Rearranging yields,

$$\|\hat{\theta} - \theta'\|_{\Sigma_{\mathcal{D}} + \frac{\alpha}{\gamma}I} \leq \frac{2\|b\|_{(\Sigma_{\mathcal{D}} + \frac{\alpha}{\gamma}I)^{-1}}}{n\gamma}.$$

The largest eigenvalue of $(\Sigma_{\mathcal{D}} + \frac{\alpha}{\gamma}I)^{-1}$ is at most $\frac{\gamma}{\alpha}$ and therefore $\|b\|_{(\Sigma_{\mathcal{D}} + \frac{\alpha}{\gamma}I)^{-1}} \leq \|b\|_2\sqrt{\frac{\gamma}{\alpha}}$. Therefore we conclude,

$$\|\hat{\theta} - \theta'\|_{\Sigma_{\mathcal{D}} + \frac{\alpha}{\gamma}I} \leq \frac{\|b\|_2}{n}\frac{1}{\sqrt{\gamma\alpha}}.$$

Standard Gaussian concentration bounds then imply that with probability at least $1 - \eta$,

$$\|\hat{\theta} - \theta'\|_{\Sigma_{\mathcal{D}} + \frac{\alpha}{\gamma}I} \leq \frac{\sigma}{n}\sqrt{\frac{2d\gamma\log\left(\frac{2}{\eta}\right)}{\alpha}}. \tag{13}$$

**An upper bound for $\|\widetilde{\theta}_{\mathbf{MLE}} - \hat{\theta}\|$.** For $w$ defined as in Algorithm 1, the operator norm bound of Lemma A.3 implies

$$\|\widetilde{\theta}_{\mathrm{MLE}} - \hat{\theta}\|_{\Sigma_{\mathcal{D}} + \lambda I} = \|w\|_{\Sigma_{\mathcal{D}} + \lambda I} \leq (4L^2 + \lambda)\|w\|_2.$$

Again standard Guassian concentration bounds imply that with probability at least $1 - \eta$,

$$\|\widetilde{\theta}_{\mathrm{MLE}} - \hat{\theta}\|_{\Sigma_{\mathcal{D}} + \lambda I} \leq (4L^2 + \lambda)\nu\sqrt{2d\log(2/\eta)}. \tag{14}$$

**Putting it all together.** Observe that by our choice of $\lambda$ and $\alpha$ we have that $\lambda \leq \frac{\alpha}{\gamma}$. Hence $\|v\|_{\Sigma_\mathcal{D}+\lambda I} \leq \|v\|_{\Sigma_\mathcal{D}+\frac{\alpha}{\gamma}I}$ for all $v \in \mathbb{R}^d$. The result now follows by applying the triangle inequality to (12), (13), and (14), applying Lemma A.8 to upper bound $\|\cdot\|_{\Sigma_\mathcal{D}+\lambda I}$ by $\|\cdot\|_{\widetilde{\Sigma}_\mathcal{D}+\lambda I}$, and plugging in the values for $\alpha$, $\beta$, $\nu$, and $\sigma$ from Algorithm 1. $\qquad\square$

## A.5 PESSIMISTIC POLICY OPTIMIZATION

We now utilize the bounds proved earlier in this section on the estimation error of Algorithm 1 and Algorithm 3 in order to complete the proof of Theorem 4.2.

*Proof of Theorem 4.2.* Let $\lambda = C\frac{\sqrt{d\log(1/\eta)\log(1/\delta)}}{\epsilon n}$. By Lemma 3.1 in Zhu et al. (2023) we have that with probability $1 - \eta$,

$$\|\hat{\theta}_{\text{MLE}} - \theta^*\|_{\Sigma_\mathcal{D}+\lambda I} \leq O\left(\sqrt{\frac{d + \log(1/\eta)}{n}} + \lambda\right).$$

Thus, by Lemma A.11, Lemma A.8, and the triangle inequality, we have that with probability $1 - \eta$

$$\|\theta^* - \widetilde{\theta}_{\text{MLE}}\|_{\widetilde{\Sigma}_\mathcal{D}+\lambda I} \leq F(n, d, \eta, \epsilon, \delta) \tag{15}$$

where

$$F(n, d, \eta, \epsilon, \delta) = O\left(\sqrt{\frac{d}{n}} + \frac{(d\log(1/\eta)\log(1/\delta))^{1/4}}{\sqrt{\epsilon n}}\right).$$

Recalling the notation $\Theta(\widetilde{\theta}_{\text{MLE}}, \lambda)$ from (2), this implies that $\theta^* \in \Theta(\widetilde{\theta}_{\text{MLE}}, \lambda)$.

Next define $J^*(\pi) = \mathbb{E}_{s\sim\rho}[\langle\theta^*, \phi(s, \pi(s))\rangle]$ and $J'(\pi) = J^*(\pi) - \langle\theta^*, v\rangle$. Let $\pi^* = \arg\min_\pi J^*(\pi)$. Note that by optimality of $\hat{\pi}_{\text{PE}}$ we have

$$\hat{J}(\hat{\pi}_{\text{PE}}) \leq \hat{J}(\pi^*) \tag{16}$$

Since $\theta^* \in \Theta(\widetilde{\theta}_{\text{MLE}}, \lambda)$ with probability $1 - \eta$, we have

$$\hat{J}(\hat{\pi}_{\text{PE}}) - J'(\hat{\pi}_{\text{PE}}) = \min_{\theta\in\Theta(\widetilde{\theta}_{\text{MLE}},\lambda)} \mathbb{E}_{s\sim\rho}[\langle\theta, \phi(s, \hat{\pi}_{\text{PE}}(s)) - v\rangle] - \mathbb{E}_{s\sim\rho}[\langle\theta^*, \phi(s, \hat{\pi}_{\text{PE}}(s)) - v\rangle]$$

$$\leq 0. \tag{17}$$

Then we can decompose the suboptimality for the output $\hat{\pi}_{\text{PE}}$ of Algorithm 3 as follows,

$$\text{SubOpt}(\hat{\pi}_{\text{PE}}) = J^*(\pi^*) - J^*(\hat{\pi}_{\text{PE}})$$
$$= J'(\pi^*) - J'(\hat{\pi}_{\text{PE}})$$
$$= (J'(\pi^*) - \hat{J}(\pi^*)) + (\hat{J}(\pi^*) - \hat{J}(\hat{\pi}_{\text{PE}})) + (\hat{J}(\hat{\pi}_{\text{PE}}) - J'(\hat{\pi}_{\text{PE}}))$$

By (16) and (17) the latter two differences above are less than zero, hence

$$\text{SubOpt}(\hat{\pi}_{\text{PE}}) \leq J'(\pi^*) - \hat{J}(\pi^*)$$
$$= \sup_{\theta\in\Theta(\widetilde{\theta}_{\text{MLE}},\lambda)} \mathbb{E}_{s\sim\rho}[\langle\theta^* - \theta, \phi(s, \pi^*(s)) - v\rangle]$$
$$= \mathbb{E}_{s\sim\rho}[\langle\theta^* - \widetilde{\theta}_{\text{MLE}}, \phi(s, \pi^*(s)) - v\rangle] + \sup_{\theta\in\Theta(\widetilde{\theta}_{\text{MLE}},\lambda)} \mathbb{E}_{s\sim\rho}[\langle\widetilde{\theta}_{\text{MLE}} - \theta, \phi(s, \pi^*(s)) - v\rangle] \tag{18}$$

By construction we have that for all $\theta \in \Theta(\widetilde{\theta}_{\text{MLE}}, \lambda)$ the Cauchy-Schwarz inequality implies

$$\mathbb{E}_{s\sim\rho}\left[\langle\widetilde{\theta}_{\text{MLE}} - \theta, \phi(s, \pi^*(s)) - v\rangle\right] \leq \|\widetilde{\theta}_{\text{MLE}} - \theta\|_{\widetilde{\Sigma}_\mathcal{D}+\lambda I}\|(\widetilde{\Sigma}_\mathcal{D} + \lambda I)^{-1/2}(\phi(s, \pi^*(s)) - v)\|_2$$
$$\leq F(n, d, \eta, \epsilon, \delta) \cdot \|(\widetilde{\Sigma}_\mathcal{D} + \lambda I)^{-1/2}(\phi(s, \pi^*(s)) - v)\|_2$$

As $\theta^* \in \Theta(\widetilde{\theta}_{\text{MLE}}, \lambda)$ with probability $1 - \eta$, we have that both terms in (18) take the form $\mathbb{E}_{s\sim\rho}\left[\langle\widetilde{\theta}_{\text{MLE}} - \theta, \phi(s, \pi^*(s)) - v\rangle\right]$ for some $\theta \in \Theta(\widetilde{\theta}_{\text{MLE}}, \lambda)$. Finally, substituting $2\lambda$ for $\lambda$ and applying Lemma A.9 implies the desired result. $\qquad\square$

## B  PROOFS FOR CONTEXTUAL BANDITS WITH $K$-WISE COMPARISONS

We begin, as in the pairwise case, with some basic properties of the loss and covariance in the $K$-wise setting.

### B.1  BASIC PROPERTIES OF $\ell_{\mathcal{D}_K}$ AND $\Sigma_{\mathcal{D}_K}$

The loss for the $K$-wise Plackett-Luce model is given by

$$\ell_{\mathcal{D}_K}(\theta) = -\frac{1}{n} \sum_{i=1}^{n} \sum_{j=1}^{K} \log \left( \frac{\exp\left(\left\langle \theta, \phi(s^i, a^i_{\sigma_i(j)})\right\rangle\right)}{\sum_{k=j}^{K} \exp\left(\left\langle \theta, \phi(s^i, a^i_{\sigma_i(k)})\right\rangle\right)} \right).$$

We will use the following notation throughout this section,

$$x^i_{jk} = \phi(s^i, a^i_{\sigma_i(j)}) - \phi(s^i, a^i_{\sigma_i(k)}).$$

The gradient and Hessian of $\ell_{\mathcal{D}_K}$ are given by the following formulas.

*Claim* B.1.

$$\nabla \ell_{\mathcal{D}_K}(\theta) = -\frac{1}{n} \sum_{i=1}^{n} \sum_{j=1}^{K} \sum_{k=j}^{K} \frac{\exp\left(\left\langle \theta, \phi(s^i, a^i_{\sigma_i(j)})\right\rangle\right)}{\sum_{l=j}^{K} \exp\left(\left\langle \theta, \phi(s^i, a^i_{\sigma_i(l)})\right\rangle\right)} \cdot x^i_{jk}.$$

*Claim* B.2.

$$\nabla^2 \ell_{\mathcal{D}_K}(\theta) = \frac{1}{n} \sum_{i=1}^{n} \sum_{j=1}^{K} \sum_{k=j}^{K} \sum_{l=j}^{K} \frac{\exp\left(\left\langle \theta, \phi(s^i, a^i_{\sigma_i(j)})\right\rangle\right)}{\sum_{l=j}^{K} \exp\left(\left\langle \theta, \phi(s^i, a^i_{\sigma_i(l)})\right\rangle\right)} \cdot x^i_{kl} x^{i\top}_{kl}.$$

These formulas lead directly to an upper bound on the norm of the gradient and the operator norm of the Hessian of $\ell_{\mathcal{D}_K}$.

**Lemma B.3.** *For all $\theta$,*

1. $\|\nabla \ell_{\mathcal{D}_K}(\theta)\|_2 \leq 2K^2 L$

2. $\|\nabla^2 \ell_{\mathcal{D}_K}(\theta)\|_{op} \leq 4K^3 L^2$

*Proof.* Observe first that $\|x_i\|_2 \leq 2L$ because $\|\phi(s, a)\| \leq L$. By Claim B.1, the gradient $\nabla \ell_{\mathcal{D}_K}(\theta)$ is the average of $n$ sums of $K^2$ vectors each of length at most $2L$. Similarly by Claim B.2, $\nabla^2 \ell_{\mathcal{D}_K}(\theta)$ is the average of $n$ sums of $K^3$ rank-one matrices, each of operator norm at most $\|x_i\|_2^2 \leq 4L^2$. □

The proof Theorem 4.1 in Zhu et al. (2023) implies that for all $\theta \in \Theta_B$ and $v \in \mathbb{R}^d$

$$v^\top \nabla^2 \ell_{\mathcal{D}_K}(\theta) v \geq \gamma_K v^\top \Sigma_{\mathcal{D}_K} v = \gamma_K \|v\|^2_{\Sigma_{\mathcal{D}_K}}. \tag{19}$$

where $\gamma_K = \frac{1}{2} \exp(-4LB)$. In particular, we have the following lemma,

**Lemma B.4.** *$\ell_{\mathcal{D}_K}$ is strongly convex on the set $\Theta_B$ with respect to the semi-norm $\|\cdot\|_{\Sigma_{\mathcal{D}_K}}$. That is, there exists a constant $\gamma_K > 0$ such that,*

$$\ell_{\mathcal{D}_K}(\theta + \Delta) - \ell_{\mathcal{D}_K}(\theta) - \langle \nabla \ell_{\mathcal{D}}(\theta), \theta \rangle \geq \frac{\gamma_K}{2} \|\Delta\|^2_{\Sigma_{\mathcal{D}_K}} \tag{20}$$

*for all $\theta \in \Theta_B$, and $\Delta$ such that $(\theta + \Delta) \in \Theta_B$.*

### B.2  PRIVATE COVARIANCE FOR $K$-WISE COMPARISONS

We obtain privacy for the feature covariance matrix $\Sigma_{\mathcal{D}_K}$ via the Gaussian mechanism. The main point is use Algorithm 2 with the variance of the Gaussian mechanism increased by a constant factor depending only on $K$.

**Lemma B.5.** *Let $\sigma^2 = \frac{64\log(\frac{1}{\delta})K^6 L^4}{\epsilon^2 n^2}$ and $G \sim \mathcal{N}(0, \sigma^2)^{d \times d}$. Then $\widetilde{\Sigma}_{\mathcal{D}_K} = \Sigma_{\mathcal{D}_K} + G$ is $(\epsilon/2, \delta/2)$-differentially private.*

*Proof.* For a dataset $\mathcal{D}'_K$ differing in one query $(s, a_1, \dots a_K, \sigma)$ from $\mathcal{D}_K$ we have

$$\|\Sigma_{\mathcal{D}_K} - \Sigma_{\mathcal{D}'_k}\|_2 \leq \frac{1}{n}K^3 \|x^i_{kl} x^{i\top}_{kl}\|_2 = \frac{1}{n}K^3 \|x^i_{kl}\|_2^2 \leq \frac{4K^3 L^2}{n}.$$

The standard analysis of the Gaussian mechanism (Dwork and Roth, 2014) then implies that $\widetilde{\Sigma}_{\mathcal{D}_K}$ is $(\epsilon/2, \delta/2)$-differentially private when setting $\sigma^2 = \frac{64\log(\frac{1}{\delta})K^6 L^4}{\epsilon^2 n^2}$. $\qquad\square$

### B.3 PRIVACY OF OBJECTIVE-PERTURBED MLE FOR $K$-WISE COMPARISONS

**Lemma B.6.** *Algorithm 1 applied to $\ell_{\mathcal{D}_K}$ and $\mathcal{D}_K$ satisfies $(\epsilon/2, \delta/2)$-differential privacy, when $\alpha$ is adjusted by a constant factor.*

*Proof.* First define

$$\ell^i_{\mathcal{D}_K}(\theta) = \sum_{j=1}^{K} \log \left( \frac{\exp\left(\left\langle \theta, \phi(s^i, a^i_{\sigma_i(j)}) \right\rangle\right)}{\sum_{k=j}^{K} \exp\left(\left\langle \theta, \phi(s^i, a^i_{\sigma_i(k)}) \right\rangle\right)} \right).$$

As pointed out in the discussion after Theorem 5.6 Bassily et al. (2019b), the analysis of objective perturbation by Iyengar et al. (2019) implies that one can still achieve differential privacy when the rank of $\nabla^2 \ell^i_{\mathcal{D}_K}(\theta)$ is larger than one. In particular, by Claim B.2,

$$\nabla^2 \ell^i_{\mathcal{D}_K}(\theta) = \sum_{j=1}^{K}\sum_{k=j}^{K}\sum_{l=j}^{K} \frac{\exp\left(\left\langle \theta, \phi(s^i, a^i_{\sigma_i(j)}) \right\rangle\right)}{\sum_{l=j}^{K} \exp\left(\left\langle \theta, \phi(s^i, a^i_{\sigma_i(l)}) \right\rangle\right)} \cdot x^i_{kl} x^{i\top}_{kl},$$

which evidently has rank at most $K^3$. Thus the analysis of Iyengar et al. (2019) implies that we need only increase $\alpha$ by a constant factor (depending only on $K$) in order to achieve $(\epsilon, \delta)$-differential privacy. $\qquad\square$

We now can conclude with our main privacy theorem for $K$-wise comparisons.

*Proof of Theorem 4.3.* $\widetilde{\theta}_{\mathrm{MLE}_K}$ is $(\epsilon/2, \delta/2)$-differentially private by Lemma B.6, and $\widetilde{\Sigma}_{\mathcal{D}_K}$ is $(\epsilon/2, \delta/2)$-differentially private by Lemma B.5. Thus, standard composition implies that the pair $(\widetilde{\theta}_{\mathrm{MLE}_K}, \widetilde{\Sigma}_{\mathcal{D}_K})$ is $(\epsilon, \delta)$-differentially private. $\qquad\square$

### B.4 APPROXIMATION ERROR AND PESSIMISTIC POLICY OPTIMIZATION FOR $K$-WISE COMPARISONS

At this point, one can check that the proofs of Lemma A.8 and Lemma A.9, as well as those of all the results in Section A.4 and Section A.5 go through, with the only change being an adjustment of the parameters by constant factors depending only on $K$. Thus, following these proofs with $\Sigma_{\mathcal{D}_K}$ substituted for $\Sigma_{\mathcal{D}}$ and $\hat{\theta}_{\mathrm{MLE}_K}$ substituted for $\hat{\theta}_{\mathrm{MLE}}$ yields Theorem 4.4.

## C PROOFS FOR GENERAL MDPS

### C.1 BASIC PROPERTIES OF $\ell_{\mathcal{D}_\tau}$ AND $\Sigma_{\mathcal{D}_\tau}$

For each tuple $(\tau^i_1, \tau^i_0, y^i) \in \mathcal{D}_\tau$ we denote the two sequences of states and actions by $\tau^i_1 = (s^i_{h1}, a^i_{h1})^H_{h=1}$ and $\tau^i_0 = (s^i_{h0}, a^i_{h0})^H_{h=1}$. The loss for general MDPs is given by the log likelihood of

the Bradley-Terry-Luce model applied to trajectory comparisons,

$$\ell_{\mathcal{D}_\tau}(\theta) = -\frac{1}{n}\sum_{i=1}^{n}\log\left(\mathbf{1}[y_i = 1]\frac{\exp\left(\sum_{h=1}^{H} r_{\theta^*}(s_{h1}^i, a_{h1}^i)\right)}{\exp\left(\sum_{h=1}^{H} r_{\theta^*}(s_{h0}^i, a_{h0}^i)\right) + \exp\left(\sum_{h=1}^{H} r_{\theta^*}(s_{h1}^i, a_{h1}^i)\right)}\right.$$

$$\left. +\mathbf{1}[y_i = 0]\frac{\exp\left(\sum_{h=1}^{H} r_{\theta^*}(s_{h0}^i, a_{h0}^i)\right)}{\exp\left(\sum_{h=1}^{H} r_{\theta^*}(s_{h0}^i, a_{h0}^i)\right) + \exp\left(\sum_{h=1}^{H} r_{\theta^*}(s_{h1}^i, a_{h1}^i)\right)}\right).$$

We will use the following notation throughout this section,

$$x_i = \sum_{h=1}^{H}\phi(s_{h1}^i, a_{h1}^i) - \phi(s_{h0}^i, a_{h0}^i).$$

The gradient and Hessian of $\ell_{\mathcal{D}_\tau}$ are given by the following formulas.

*Claim* C.1.

$$\nabla\ell_{\mathcal{D}_\tau}(\theta) = -\frac{1}{n}\sum_{i=1}^{n}\left(\mathbf{1}[y_i = 1]\frac{\exp(-\langle\theta, x_i\rangle)}{1 + \exp(-\langle\theta, x_i\rangle)} - \mathbf{1}[y_i = 0]\frac{1}{1 + \exp(-\langle\theta, x_i\rangle)}\right)x_i$$

*Claim* C.2.

$$\nabla^2\ell_{\mathcal{D}_\tau}(\theta) = \frac{1}{n}\sum_{i=1}^{n}\frac{\exp(-\langle\theta, x_i\rangle)}{(1 + \exp(-\langle\theta, x_i\rangle))^2}x_i x_i^\top$$

These formulas lead directly to an upper bound on the norm of the gradient and the operator norm of the Hessian of $\ell_{\mathcal{D}_\tau}$.

**Lemma C.3.** *For all $\theta$,*

1. $\|\nabla\ell_{\mathcal{D}_\tau}(\theta)\|_2 \le 2HL$

2. $\|\nabla^2\ell_{\mathcal{D}_\tau}(\theta)\|_{op} \le 4H^2L^2$

*Proof.* Observe first that $\|x_i\|_2 \le 2HL$ because it is the sum of $H$ vectors each of norm at most $2\|\phi(s,a)\| \le 2L$. By Claim C.1, the gradient $\nabla\ell_{\mathcal{D}_\tau}(\theta)$ is the average of $n$ vectors each of length at most $2HL$. Similarly by Claim C.2, $\nabla^2\ell_{\mathcal{D}_\tau}(\theta)$ is the average of $n$ vectors, each of operator norm at most $\|x_i\|_2^2 \le 4H^2L^2$. $\qquad\square$

The proof Lemma 5.1 in Zhu et al. (2023) implies that for all $\theta \in \Theta_B$ and $v \in \mathbb{R}^d$

$$v^\top\nabla^2\ell_{\mathcal{D}_\tau}(\theta)v \ge \gamma_\tau v^\top\Sigma_{\mathcal{D}_\tau}v = \gamma_\tau\|v\|_{\Sigma_{\mathcal{D}_\tau}}^2. \qquad (21)$$

where $\gamma_\tau = \frac{1}{2+\exp(-2HLB)+\exp(2HLB)}$. In particular, we have the following lemma,

**Lemma C.4.** *$\ell_{\mathcal{D}_\tau}$ is strongly convex on the set $\Theta_B$ with respect to the semi-norm $\|\cdot\|_{\Sigma_{\mathcal{D}_\tau}}$. That is, there exists a constant $\gamma_\tau > 0$ such that,*

$$\ell_{\mathcal{D}}(\theta + \Delta) - \ell_{\mathcal{D}_\tau}(\theta) - \langle\nabla\ell_{\mathcal{D}_\tau}(\theta), \theta\rangle \ge \frac{\gamma_\tau}{2}\|\Delta\|_{\Sigma_{\mathcal{D}}}^2 \qquad (22)$$

*for all $\theta \in \Theta_B$, and $\Delta$ such that $(\theta + \Delta) \in \Theta_B$.*

## C.2 PRIVATE COVARIANCE FOR GENERAL MDPS

We obtain privacy for the feature covariance matrix $\Sigma_{\mathcal{D}_\tau}$ via the Gaussian mechanism. The main point is to use Algorithm 2 with the variance of the Gaussian mechanism increased by a constant factor depending only on $H$.

**Lemma C.5.** *Let $\sigma^2 = \frac{64\log(\frac{1}{\delta})H^4L^4}{\epsilon^2 n^2}$ and $G \sim \mathcal{N}(0, \sigma^2)^{d\times d}$. Then $\widetilde{\Sigma}_{\mathcal{D}_\tau} = \Sigma_{\mathcal{D}_\tau} + G$ is $(\epsilon/2, \delta/2)$-differentially private.*

*Proof.* For a dataset $\mathcal{D}'_\tau$ differing in one query $(s, a_1, \ldots a_K, \sigma)$ from $\mathcal{D}_\tau$ we have

$$\|\Sigma_{\mathcal{D}_K} - \Sigma_{\mathcal{D}'_\tau}\|_2 \leq \frac{1}{n}\|x_i x_i^\top\|_2 = \frac{1}{n}\|x^i\|_2^2 \leq \frac{4H^2 L^2}{n}.$$

The standard analysis of the Gaussian mechanism (Dwork and Roth, 2014) then implies that $\Sigma_{\mathcal{D}_\tau}$ is $(\epsilon/2, \delta/2)$-differentially private when setting $\sigma^2 = \frac{64 \log(\frac{1}{\delta}) H^4 L^4}{\epsilon^2 n^2}$. $\quad\square$

### C.3 PRIVACY OF OBJECTIVE-PERTURBED MLE FOR GENERAL MDPS

**Lemma C.6.** *Algorithm 1 applied to $\ell_{\mathcal{D}_\tau}$ and $\mathcal{D}_\tau$ satisfies $(\epsilon/2, \delta/2)$-differential privacy, when the input parameters are adjusted by at most a constant factor depending only on $H$.*

*Proof.* Similarly to the case of pairwise comparisons for contextual bandits in Lemma C.6, the Hessian $\nabla^2 \ell_{\mathcal{D}_\tau}(\theta)$ is the sum of $n$ rank-one terms. Thus, after adjusting the parameters by a constant factor depending on $H$, Theorem 5.6 of Bassily et al. (2019b) implies that $\widetilde{\theta}_{\mathrm{MLE}_\tau}$ is $(\epsilon/2, \delta/2)$-differentially private. $\quad\square$

We now can conclude with our main privacy theorem for the general MDP setting.

*Proof of Theorem 5.1.* $\widetilde{\theta}_{\mathrm{MLE}_\tau}$ is $(\epsilon/2, \delta/2)$-differentially private by Lemma C.6, and $\widetilde{\Sigma}_{\mathcal{D}_\tau}$ is $(\epsilon/2, \delta/2)$-differentially private by Lemma C.5. Thus, standard composition implies that the pair $(\widetilde{\theta}_{\mathrm{MLE}_\tau}, \widetilde{\Sigma}_{\mathcal{D}_\tau})$ is $(\epsilon, \delta)$-differentially private. $\quad\square$

### C.4 APPROXIMATION ERROR AND PESSIMISTIC POLICY OPTIMIZATION FOR GENERAL MDPS

As in the case of $K$-wise comparisons, the proofs of Lemma A.8 and Lemma A.9, as well as those of all the results in Section A.4 and Section A.5 go through, with the only change being an adjustment of the parameters by constant factors depending only on $H$, $L$, and $B$. The only additional modification necessary for the general MDP setting is to use the policy-dependent distribution on states and actions $\rho_\pi$ in the place of the fixed distribution on states $\rho$ in the proof from Section A.5. Thus, following these proofs with $\Sigma_{\mathcal{D}_\tau}$ substituted for $\Sigma_{\mathcal{D}}$ and $\hat{\theta}_{\mathrm{MLE}_\tau}$ substituted for $\hat{\theta}_{\mathrm{MLE}}$ yields Theorem 5.2.

