# OpenReview forum: "Aligning With Human Values Without Revealing Human Judgements"
_ICLR.cc/2025/Conference — Submitted to ICLR 2025_

### Official Review · Reviewer_Nm6F · 2024-10-27

**Soundness:** 2
**Presentation:** 2
**Contribution:** 2
**Rating:** 3
**Confidence:** 4

**Summary:**

The paper investigated RHLF from K-wise comparisons under central differential privacy. The authors designed private algorithms for the pessimistic method and provided the upper bounds for the sub-optimality gap.

**Strengths:**

The paper gives some theoretical results for RLHF under DP.

**Weaknesses:**

1. The literature review is not comprehensive enough, see Questions part for more details.
2. There are some typos in important equations and some concepts are not clear, see Questions part for more details.
3. There are no lower bounds and experiments.

**Questions:**

1. In lines 110-111, the notation is confusing for $\sum_{j=1}^K \sum_{j=k+1}^K$. There are two "j" indexes.
2. The authors claim that the paper provides the first algorithms that can achieve provable privacy of human judgments. However, [1] studied RLHF under both local DP and central DP. And the authors didn't cite the paper and didn't compare the results with theirs.
3. The concept of DP in section 2.1 is not clear for the corresponding case of RLHF. What is the output of algorithm $\mathcal{A}$? Why did you choose approximate DP, not pure DP? Why did you choose central DP, not local DP?
4. The authors should cite earlier work like [2] for the pessimistic method.
5. In algorithm 1, why did you use objective perturbation in Line 4 and output perturbation for DP together? See section 5.2 in [1] for objective perturbation to guarantee DP. Why did you choose the regularizer parameter $\alpha$ to be related to private parameter $\epsilon$.
6. What is the dependence on $B$ and $L$ in your final suboptimality gap?

[1]. Chowdhury, Sayak Ray, Xingyu Zhou, and Nagarajan Natarajan. "Differentially private reward estimation with preference feedback." International Conference on Artificial Intelligence and Statistics. PMLR, 2024.

[2]. Ying Jin, Zhuoran Yang, and Zhaoran Wang. Is pessimism provably efficient for offline rl? In International Conference
on Machine Learning, pages 5084–5096. PMLR, 2021

---

> ### Author Response · Authors · 2024-11-24
> **Author Response**
>
> ---
>
> *1. "The authors claim that the paper provides the first algorithms that can achieve provable privacy of human judgments. However, [1] studied RLHF under both local DP and central DP. And the authors didn't cite the paper and didn't compare the results with theirs.'*
>
> ---
>
> Please see our response to Reviewer CYVZ above. Note also that our results were online before [1].
>
> [1] Sayak Ray Chowdhury and Xingyu Zhou and Nagarajan Natarajan. Differentially private reward estimation with preference feedback, AISTATS 2024.
>
> ---
>
> *2. "The concept of DP in section 2.1 is not clear for the corresponding case of RLHF. What is the output of algorithm $\mathcal{A}$? Why did you choose approximate DP, not pure DP? Why did you choose central DP, not local DP?"*
>
> ---
>
> In the full RLHF pipeline the algorithm $\mathcal{A}$ outputs a policy $\pi$ which assigns a probability $\pi(s,a)$ to each state-action pair. Thus $\pi \in \mathbb{R}^{S\times A}$. The minimax optimal rates achievable with approximate differential privacy are provably not achievable with pure DP, even in the simpler setting of private empirical risk minimization [1].
> Hence, we chose approximate DP, as is standard in the literature on private convex optimization and private ERM. Central DP makes the most sense in the context of RLHF training where raters interact with an LLM hosted by a trusted provider (all of the strongest frontier models are hosted in this way). Thus, if these raters’ responses are subsequently used by the trusted provider to train a new LLM via RLHF, then the rater’s private data could be leaked to anyone who interacts with the new LLM. This corresponds precisely to the setting of central DP.
>
> [1] Bassily, Raef, Adam Smith, and Abhradeep Thakurta. Private empirical risk minimization: Efficient algorithms and tight error bounds. FOCS, 2014.
>
> ---
>
> *3. "In algorithm 1, why did you use objective perturbation in Line 4 and output perturbation for DP together? See section 5.2 in [1] for objective perturbation to guarantee DP. Why did you choose the regularizer parameter $\alpha$ to be related to private parameter $\epsilon$."*
>
> ---
>
> Objective perturbation alone only guarantees that the true optimum of the perturbed objective is differentially private. However, the true optimum is never exactly achieved via first-order methods, but rather an approximate optimum is computed. In this case, output perturbation is additionally necessary in order to maintain privacy.
>
> The relationship between $\alpha$ and the privacy parameter arises due to the need to trade-off errors in the $\ell_2$-norm and the dataset-dependent norm $\lVert\cdot\rVert\_{\Sigma\_{\mathcal{D}} + \lambda I}$.
>
> ---
>
> *4. "The authors should cite earlier work like [2] for the pessimistic method."*
>
> ---
>
> Thank you for providing the reference. We can add the citation to [2].
>
> ---
>
> *5. ”What is the dependence on $B$ and $L$ in your final suboptimality gap?”*
>
> ---
>
> The dependence on $B$ and $L$ is $O(L^2 + B)$.
>
> ---
>
> *6. "In lines 110-111, the notation is confusing for $\sum_{j=1}^K \sum_{j=k+1}^K$. There are two "j" indexes."*
>
> ---
>
> Thank you for catching this typo, it should instead read $\sum_{j=1}^K \sum_{k=j+1}^K$.

---

### Official Review · Reviewer_QAVq · 2024-10-28

**Soundness:** 3
**Presentation:** 3
**Contribution:** 3
**Rating:** 6
**Confidence:** 2

**Summary:**

The paper proposes a method for aligning large language models with human values while satisfying differential privacy requirements. The authors achieve minimax-optimal performance, ensuring the security of private data. Their results demonstrate that even under privacy constraints ($(\epsilon,\delta)$-DP), model alignment with human values is effective in contextual bandits, maintaining performance within $O(\sqrt{d/n})$ of the optimal policy when compared with non-private methods. This approach is then extended to general MDPs, where human preferences are collected over trajectory pairs. The algorithm preserves privacy while matching the performance of non-private methods.

**Strengths:**

Please find the strengths below:
1. The paper introduces differential privacy to model alignment while preserving similar performance with respect to $d$ and $n$.
2. The paper is not limited to $K=2$ but also provides results for general $K$, which is closer to the setting of RLHF.
3. The results are also extended to general MDPs, matching previous performance levels without satisfying differential privacy constraints.

**Weaknesses:**

Please find the weaknesses below:
1. The paper does not include any numerical experiments or experiments on real datasets. While the theoretical results for both contextual bandits and MDPs are asymptotic to the optimal policy, it would be valuable to observe the overhead of differential privacy in practice, especially when $d$ is small.
2. Although the performance matches that of methods without differential privacy in both contextual bandits and MDPs settings, the paper does not provide a lower bound. It would be beneficial to include a lower bound with respect to $\epsilon$ and $\delta$.
3. The paper is limited to the Plackett-Luce model; it would be interesting to explore whether similar results could be derived for other comparison models, such as the Thurstone-Mosteller model.

**Questions:**

Refer to the weaknesses section above, and find the additional questions below:
1. In Section 4.2, the paper discusses the case of $k$-wise comparison. Is it possible for agents to make comparisons involving different numbers of items?
2. Is $O(\sqrt{d/n})$ optimal for both contextual bandits and MDP settings with and without differential privacy? If so, it would be helpful to mention these results in the main paper.
3. In the MDP setting in Section 5, is there a specific reason why the results focus on $K=2$ rather than a general $K$?

**Details Of Ethics Concerns:**

I did not identify any ethical concerns.

---

> ### Author Response · Authors · 2024-11-24
> **Author Response**
>
> Thank you for stating that our paper achieves minimax-optimal performance, ensuring the security of private data and further pointing out that our algorithm preserves privacy while matching the performance of non-private methods.
>
> ---
>
> *1. ”In Section 4.2, the paper discusses the case of $k$-wise comparison. Is it possible for agents to make comparisons involving different numbers of items?”*
>
> ---
>
> A standard approach to deal with situations where different numbers of items are compared across raters is to interpret each ranking of $K$-items as $\binom{K}{2}$ pairwise comparisons, and then to simply use the Bradley-Terry model for this new dataset of pairwise comparisons.
>
> ---
>
> *2. ”Is $O(\sqrt{d/n})$ optimal for both contextual bandits and MDP settings with and without differential privacy? If so, it would be helpful to mention these results in the main paper.”*
>
> ---
>
> Yes, indeed $O(\sqrt{d/n})$ is minimax optimal for the contextual bandits setting, and hence, because contextual bandits are a subset of MDPs, is also optimal for the MDP setting. We can add a mention of this to the main paper.
>
> ---
>
> *3. ”In the MDP setting in Section 5, is there a specific reason why the results focus on $K=2$ rather than a general $K$?”*
>
> ---
>
> This is primarily for clarity of the presentation, as larger $K$ leads to very unwieldy notation without much additional insight. A generalization to larger $K$ can be obtained via a straightforward translation of our results for $K$-wise comparisons to the case of trajectories.
>
> ---
>
> *4. ”The paper is limited to the Plackett-Luce model; it would be interesting to explore whether similar results could be derived for other comparison models, such as the Thurstone-Mosteller model.”*
>
> ---
>
> The reason to focus on the Plackett-Luce model and the pairwise special case of the Bradley-Terry model is that this is the standard model used in practice for RLHF on LLMs. Thus, in order to prove results most closely aligned with current best-practices, we chose to focus on the Plackett-Luce model.

---

### Official Review · Reviewer_eQ23 · 2024-11-02

**Soundness:** 3
**Presentation:** 3
**Contribution:** 3
**Rating:** 6
**Confidence:** 3

**Summary:**

The paper considers the problem of differentially private alignment via reinforcement learning from human feedback, motivated by the practical concern of trying to preserve privacy of the annotators preferences. The authors consider two main settings in this general framework: the first is a linear contextual bandit setting, and the second is a more general finite horizon MDP setting. The human feedback in the former setting consists of complete orderings over the different actions, assumed to have been drawn from the Plackett-Luce model, whereas the feedback in the latter setting is assumed to consist of a pairwise choice between two trajectories, assumed to have been drawn from the Bradley-Terry-Luce model, which is a special case of the Plackett-Luce model for pairwise comparisons. The objective is to first learn the underlying parameters of these two models (i.e. estimate the underlying human preferences) in a differentially private manner (from given offline data consisting of these trajectories and corresponding ranked preferences), and then use these estimated parameters to then learn a policy that achieves reward that is comparable to to the one achieved by a policy that knew the underlying preference vector exactly. In this setting, the authors show that it is possible to compute an $(\epsilon,\delta)$-differentially private estimator for the underlying preferences (as well as the data-covariance matrix) that for any constant $\epsilon$ and inverse polynomial $\delta$, is such that the pessimistic policy optimization algorithm given these estimated, private parameters as input, achieves reward that is minimax optimal. In other words, the additional loss in reward incurred due to enforcing $(\epsilon,\delta)$-DP is order-wise identical to that achieved by the non-private MLE estimators. These results follow by a fairly direct application of the Gaussian mechanism to the MLE estimation of the PL parameters, and data-covariances. It is important to note that these results only hold in the central model of differential privacy, where there is a central trusted curator that holds the non-private data.

**Strengths:**

In my opinion, the paper is quite well written and easy to follow. The problem of privacy more generally is an important area of consideration, so the research is also of practical significance. The algorithms are pretty simple which should make them applicable in practice. The results are also order wise optimal, and show that it is possible to achieve differential privacy without much asymptotic loss in reward.

**Weaknesses:**

Other than the fact that these results only hold assuming a linear reward function (which to be fair, the authors have also acknowledged), the only other concern I have is novelty. Differential privacy is not my primary area of work, but it seems to me that this result is obtained via a pretty straightforward application of the gaussian mechanism which I believe to be one of the standard methods in the DP world. That being said, I am no means an expert in this area, and will defer to other more knowledgeable reviewers judgement regarding the novelty and impact of this work.

**Questions:**

If the authors can elaborate what makes this result novel, and not a direct consequence of a straightforward application of the gaussian mechanism, that would help me evaluate this work more fairly.

Moreover, I find the last bit in the conclusion regarding the importance of privacy and how it can prevent data leaks by LLMs to be very much an oversell in this paper. The impact of this paper is limited to keeping the preferences of annotators who are involved in aligning these models, private. This by itself cannot and will not prevent data leaks from LLM responses due to the very nature of the way LLMs are trained. I would strongly prefer it if you could remove that statement/ reword it so as to not send a wrong message to the readers, especially ones who come from a more applied background and who may not be aware of how differential privacy works and what its limitations are.

---

> ### Author Response · Authors · 2024-11-24
> **Author Response**
>
> Thank you for writing that our paper shows that it is possible to achieve differential privacy without any asymptotic loss in reward while further our paper is quite well written and easy to follow in an important area of consideration, and thank you for providing a thoughtful review.
>
> ---
>
> *1. ”If the authors can elaborate what makes this result novel, and not a direct consequence of a straightforward application of the gaussian mechanism, that would help me evaluate this work more fairly.”*
>
> ---
>
> The main difficulty encountered here not present in prior work is that there are two stages to RLHF: reward learning and policy optimization. The asymptotically best performing algorithms for the second stage require the learned reward function parameterized by $\theta$ and the data covariance $\Sigma$ in order to perform pessimistic policy optimization using the learned rewards. Our approach is to use differentially private versions of both $\theta$ and $\Sigma$ as input to the pessimistic policy optimization step. Separately one can keep $\theta$ private with small $\ell_2$-norm error with objective perturbation methods, and $\Sigma$ private with the Gaussian mechanism. However, for the pessimistic policy optimization to succeed we actually care about privacy of $\theta$ with respect to the dataset-dependent norm $\lVert \cdot \rVert\_{\Sigma\_{\mathcal{D}} + \lambda I}$. Thus, we must adapt standard objective perturbation to account for this different norm. But there is yet another obstacle: to maintain privacy our pessimistic policy optimization algorithm must not even have access to the true $\Sigma\_{\mathcal{D}}$ but only to a noisy DP version. Thus, balancing the privacy of $\theta$ and $\Sigma\_{\mathcal{D}}$ with the different norms required for optimal downstream policy performance requires a somewhat delicate analysis that is specific to RLHF.
>
> ---
>
> *2. ”Moreover, I find the last bit in the conclusion regarding the importance of privacy and how it can prevent data leaks by LLMs to be very much an oversell in this paper. The impact of this paper is limited to keeping the preferences of annotators who are involved in aligning these models, private. This by itself cannot and will not prevent data leaks from LLM responses due to the very nature of the way LLMs are trained. I would strongly prefer it if you could remove that statement/ reword it so as to not send a wrong message to the readers, especially ones who come from a more applied background and who may not be aware of how differential privacy works and what its limitations are.”*
>
> ---
>
> We will happily rephrase the last part in the conclusion to reflect the fact that our methods maintain privacy for the human raters that are used to align large language models during RLHF, and not for the data that might have been seen during pre-training.

---

> > ### Comment · Reviewer_eQ23 · 2024-11-27
> >
> > I thank the authors for their response regarding my concern regarding novelty, which also seems to a common concern be shared by other reviewers. I am still not fully convinced that achieving privacy w.r.t. the Mahalanobis norm is truly that big a technical barrier to achieving this result, though I will still maintain my original weakly positive opinion about this paper, largely due to its potential practical applicability.

---

### Official Review · Reviewer_CYVZ · 2024-11-03

**Soundness:** 2
**Presentation:** 2
**Contribution:** 2
**Rating:** 3
**Confidence:** 4

**Summary:**

This paper addresses the problem of aligning language models with human values while preserving the privacy of human judgments used in training. By leveraging differential privacy (DP), the authors ensure that human preference data remains confidential, even during alignment through reinforcement learning from human feedback (RLHF). They adapt the Bradley-Terry-Luce and Plackett-Luce models to achieve privacy-preserving reward learning, which performs comparably to non-private methods.

**Strengths:**

+ The study of privacy protection in LLM alignment is important and timely

**Weaknesses:**

- Some result lacks rigor in the proof
- Some important references are missing
- The technical novelty is not clear

**Questions:**

1. My first concern is that this paper needs a careful literature review. In particular, several important related works are missing. For example, [R1] studied a similar private LLM alignment problem as the current paper, considering both the local and central model of DP. Moreover, several works on private RL are missing, including [R2-R5].

2. The current proof of privacy guarantee is not grounded (i.e., Theorem 4.1 and in particular Lemma A.10). The key issue is that the privacy proof in the original paper of Kifer et al 2012 has a gap, which has been mentioned in [R1, R6, R7]. As a result, since the current proof of Lemma A.10 directly uses the result in Kifer et al 2012, it needs a careful check. It seems to me that this issue has already been addressed in [R1].

3. The utility upper bound seems to be straightforward, given the standard techniques in private linear bandit and RL. Moreover, no lower bound is established, which makes it hard to evaluate the tightness.

4. Finally, it would be good to see some basic experiments to justify the theoretical results.





--

[R1] Chowdhury, Sayak Ray, Xingyu Zhou, and Nagarajan Natarajan. "Differentially private reward estimation with preference feedback." International Conference on Artificial Intelligence and Statistics. PMLR, 2024.

[R2] Qiao, Dan, and Yu-Xiang Wang. "Near-optimal differentially private reinforcement learning." International Conference on Artificial Intelligence and Statistics. PMLR, 2023.

[R3] Liao, Chonghua, Jiafan He, and Quanquan Gu. "Locally differentially private reinforcement learning for linear mixture markov decision processes." Asian Conference on Machine Learning. PMLR, 2023.

[R4] Chowdhury, Sayak Ray, and Xingyu Zhou. "Differentially private regret minimization in episodic markov decision processes." Proceedings of the AAAI Conference on Artificial Intelligence. Vol. 36. No. 6. 2022.

[R5] Zhou, Xingyu. "Differentially private reinforcement learning with linear function approximation." Proceedings of the ACM on Measurement and Analysis of Computing Systems 6.1 (2022): 1-27.

[R6] Agarwal, Naman, et al. "Differentially private and lazy online convex optimization." The Thirty Sixth Annual Conference on Learning Theory. PMLR, 2023.

[R7] Redberg, Rachel, Antti Koskela, and Yu-Xiang Wang. "Improving the privacy and practicality of objective perturbation for differentially private linear learners." Advances in Neural Information Processing Systems 36 (2023): 13819-13853.

---

> ### Author Response · Authors · 2024-11-24
> **Author Response**
>
> ---
>
> *1. "Literature review and the technical novelty"*
>
> ---
>
>
> **While a preliminary version of our results were already publicly accessible online before the paper you refer to [1], please find the core foundational differences of our paper described in detail below.**
>
>
> The most notable differences between our results and those in [1] is first, that our results maintain privacy of the reward model for each entire preference tuple $(s,a_0,a_1,y)$ consisting of a prompt $s$ two possible responses $a_0,a_1$ the preference label $y$, and second, that we analyze the impact of privacy downstream on the RLHF training. In contrast, the paper [1] focuses on label differential privacy for the reward model alone. That is, [1] only keeps the preference labels $y$ private, and no analysis is done on how this private preference model will in fact perform downstream for RLHF.
>
> This is an important distinction, because the best non-private RLHF algorithms in this setting utilize both the preference model and the data covariance (which depends on $s,a_0,a_1$) when optimizing the final policy, and thus the interplay between privacy for both the model and the covariance simultaneously is a core component of a full private RLHF pipeline.
>
> Thus, our work indeed more directly addresses the potential privacy problems of the true RLHF setting, and second, the true RLHF setting privacy guarantees require a delicate and novel analysis on privacy with respect to the performance of the RLHF policy.
>
> In more detail: first, in almost all use-cases the prompts themselves are produced by human raters or extracted from conversations with raters. Thus the prompts and the LLM responses may contain sensitive data about the rater. In fact, this sensitive data may be much more privacy-violating than the preference label alone, as it contains multiple sentences of text written by the rater.
>
> Second, the fact that we require the entire RLHF pipeline to be private, and privacy for the entire preference tuple, indeed requires a careful and distinct analysis of the privacy. The entire point of private learning of preferences is to then actually use the learned preferences to train a model via RLHF. The best known non-private RLHF algorithm utilizes pessimistic policy optimization which uses the full data covariance matrix, and this is indeed the setting of our analysis. Thus, we must keep the data covariance private, and ensure that the pessimistic policy optimization algorithm still achieves good performance when a private covariance matrix is used. See also the discussion on line 394-400 in our paper, on balancing the privacy of the data-covariance and the private MLE estimator with the accuracy of pessimistic policy optimization.
>
>
> **The paper you refer to [1] as an obstacle to the novelty of our paper, which became publicly available more than a month after our paper was publicly available**, seems to have added a vague sketch of how to extend the results to DP for the full RLHF pipeline in Appendix E **with no formal proof**, just handwaving in a single paragraph that the privacy of the covariance and the utility analysis can be done.
>
> [1] Sayak Ray Chowdhury and Xingyu Zhou and Nagarajan Natarajan. Differentially private reward estimation with preference feedback, AISTATS 2024.
>
> [2] Sayak Ray Chowdhury and Xingyu Zhou. Differentially Private Regret Minimization in Episodic Markov Decision Processes, AAAI 2022.
>
> [3] Xingyu Zhou. Differentially Private Reinforcement Learning with Linear Function Approximation, ACM on Measurement and Analysis of Computing Systems, 2022.
>
> ---
>
> *2. "The current proof of privacy guarantee is not grounded (i.e., Theorem 4.1 and in particular Lemma A.10). The key issue is that the privacy proof in the original paper of Kifer et al 2012 has a gap."*
>
> ---
>
> The paper in the reference you listed [2] indeed fixes this gap for objective perturbation of GLM-structured loss functions of the form $l(\theta,x;y) = f(\theta^{\top}x;y)$. The loss function in Theorem 4.1 indeed has this form, which can be seen by letting $x = \phi(x,a_1) - \phi(x,a_0)$. Thus, Theorem 3.1 of [2] can be directly used as a drop-in replacement for the Kifer et al 2012 result to resolve this gap.
>
> [2] Redberg, Rachel, Antti Koskela, and Yu-Xiang Wang. Improving the privacy and practicality of objective perturbation for differentially private linear learners. NeurIPS 2023.

---

### Meta-Review · Area_Chair_LJV9 · 2024-12-21

**Metareview:**

The main concern regarding the paper is the novelty and similarity with prior works, especially the main technical barrier the authors have to overcome to achieve privacy. Despite the practical implications of privacy-preserving alignment problems, no experiments are provided to validate the efficacy of the results. The paper also suffers from some minor typos, so another thorough revision would be helpful. In general, there is not enough interest across reviewers to strongly champion this paper. Based on these, I suggest authors submit to the next suitable theory-ML venue with all the necessary modifications. Also please consider adding experiments to support the theory.

**Additional Comments On Reviewer Discussion:**

See above.

---

### Decision · Program_Chairs · 2025-01-22

Reject